# Automated model-predictive design of synthetic promoters to control transcriptional profiles in bacteria

Travis L. LaFleur [1], Ayaan Hossain [2] & Howard M. Salis [1,2,3,4] ✉

Transcription rates are regulated by the interactions between RNA polymerase, sigma factor, and promoter DNA sequences in bacteria. However, it remains unclear how non-canonical sequence motifs collectively control transcription rates. Here, we combine massively parallel assays, biophysics, and machine learning to develop a 346-parameter model that predicts site-specific transcription initiation rates for any $\sigma^{70}$ promoter sequence, validated across 22132 bacterial promoters with diverse sequences. We apply the model to predict genetic context effects, design $\sigma^{70}$ promoters with desired transcription rates, and identify undesired promoters inside engineered genetic systems. The model provides a biophysical basis for understanding gene regulation in natural genetic systems and precise transcriptional control for engineering synthetic genetic systems.

Transcription is the gene expression process responsible for producing all RNA and is a common engineering target for creating novel products, including microbial chemical factories, toxin-sensing genetic circuits, and mRNA vaccines[1–3]. However, while DNA assembly techniques enable the construction of custom-designed genetic systems[4], it remains challenging to a priori predict and control a system's gene expression profile[5], for example, by initiating transcription with desired rates at specific DNA start sites, while minimizing transcription from all other DNA sequence regions. Currently, transcriptional control relies on empirical characterization of promoters as modular genetic parts[6]. Applying modular design to transcriptional control ignores other sources of transcription, for example, inside coding regions, as well as local and long-distance interactions that alter transcription rates and start sites in unexpected ways[7]. Poorly controlled transcriptional profiles can lead to malfunctioning genetic systems, including the undesired production of anti-sense RNA and truncated proteins as well as the misbalancing of protein expression levels that lead to lower system activities[8,9].

A key challenge is to quantitatively predict how polymerase initiation complex−RNA polymerase (RNAP) and a sigma factor (σ) in bacteria−interacts with arbitrary DNA sequences[10,11]. To address

this challenge, researchers have characterized thousands of promoters in vivo using high-throughput cloning and developed models ranging in predictability[11–17]. Despite these efforts, it remains unclear how the strengths of multiple interactions[18–26] collectively determine transcription initiation rates and start sites, particularly when bound to RNAP/σ[70], which initiates transcription at the majority of bacterial promoters. Furthermore, it remains a challenge to accurately measure the strength of these interactions while taking into account differences in mRNA decay rates[27,28], the presence of cryptic transcriptional start sites (TSSs)[29–34], and incomplete self-cleavage by insulating ribozymes[35–37].

To address these challenges, we carried out massively parallel in vitro experiments on designed promoter sequences with designed barcodes to systematically measure the interactions controlling site-specific transcription at σ[70] promoters (Fig. 1A). With this data, we developed a statistical thermodynamic model that calculates how RNAP/σ[70] interacts with arbitrary DNA to predict transcription initiation rates at each position. The model has only 346 interaction energy parameters, but accurately predicts the transcription rates of 22,132 bacterial promoters with diverse sequences. We show how the model enables the automated design and debugging of transcriptional profiles in engineered genetic systems.

[1]Department of Chemical Engineering, Pennsylvania State University, University Park, PA 16801, USA. [2]Bioinformatics and Genomics, Pennsylvania State University, University Park, PA 16801, USA. [3]Department of Biological Engineering, Pennsylvania State University, University Park, PA 16801, USA. [4]Department of Biomedical Engineering, Pennsylvania State University, University Park, PA 16801, USA. ✉e-mail: salis@psu.edu

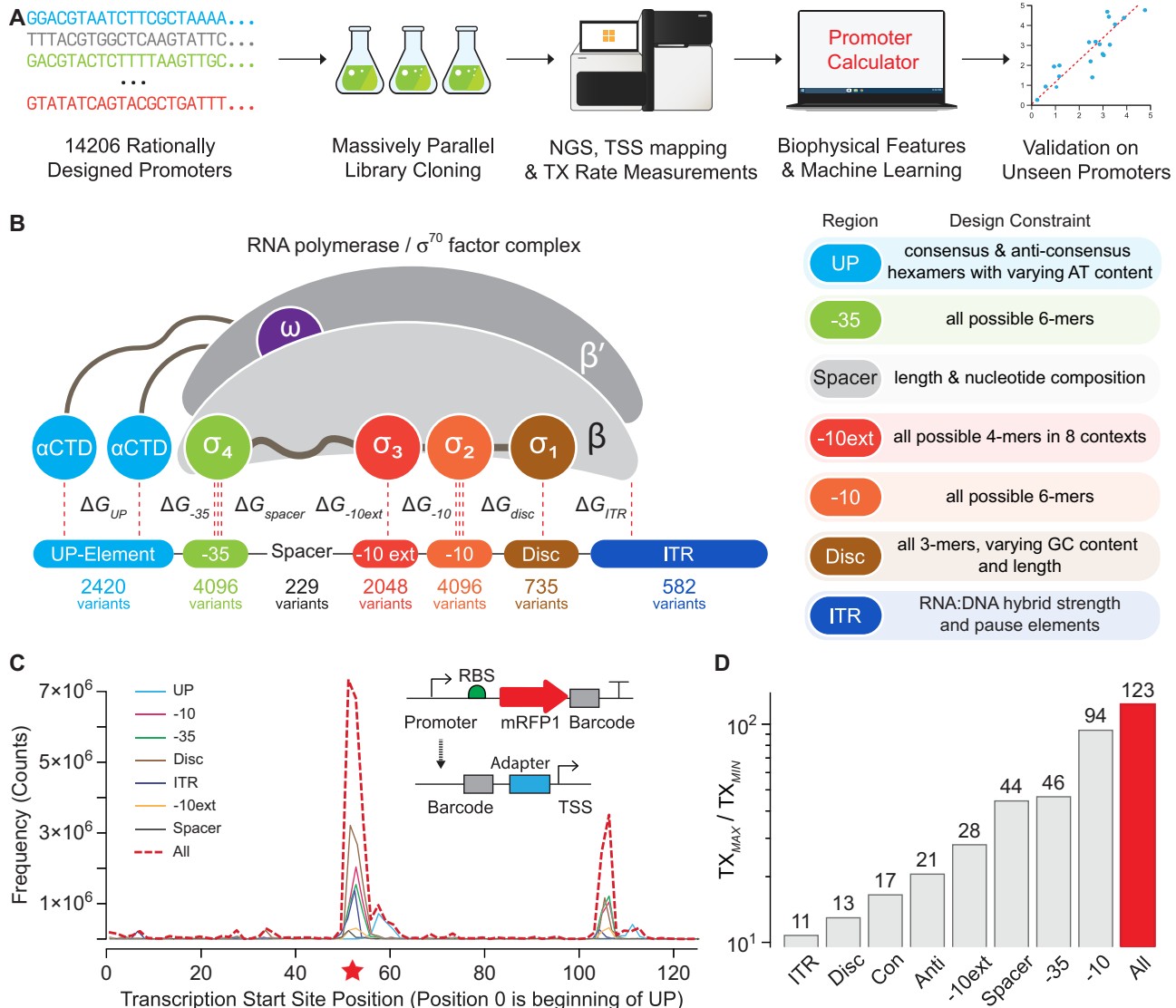

**Fig. 1 | Massively Parallel Transcription Rate and Start Site Measurements.**
**A** Model development combined promoter design, barcoded oligopool synthesis, library cloning & culturing, and next-generation sequencing to measure the transcription start site distribution and transcription rate of each promoter variant. **B** The interaction strengths between RNAP/σ[70] and promoter DNA control transcription initiation rates. 14206 promoter variants were designed to quantify how sequence modifications affect each interaction. Sequence design criteria are shown. **C** The frequencies of observed transcription start sites are shown for each set of promoter variants. The star indicates the predominant start site. The inset schematic shows the system architecture, the locations of the promoter and barcode variants, and the cDNA architecture after library preparation. **D** The transcription rates' dynamic ranges are shown for each set of promoter variants, considering only the predominant start site. Con: UP element promoter variants with consensus hexamers. Anti: UP element promoter variants with anti-consensus hexamers. Data are provided in Supplementary Data 1.

## Results

### Model formulation and library design

To begin, we designed 14,206 promoters with varied sequence motifs to systematically perturb the interactions that affect RNAP/σ[70] binding and transcriptional initiation (Fig. 1B). These interactions occur at DNA sites known by their canonical positions[38,39], including (i) an upstream 6-nucleotide site called the −35 motif; (ii) a 20-nucleotide region that appears upstream of the −35 motif, called the UP element; (iii) a downstream 6-nucleotide site called the −10 motif; (iv) a spacer region that separates the −10 and −35 motifs; (v) a 4-nucleotide site upstream of the −10 motif, called the −10 extended motif (−10 ext); (vi) a typically 6-nucleotide region in between the −10 motif and TSS, called the discriminator (Disc); and (vii) the first 20 transcribed nucleotides, called the initial transcribed region (ITR).

Initial RNAP/σ[70] binding to a promoter is controlled by the interaction Gibbs free energies (ΔG) at the UP, −35, −10 extended,

and −10 motifs as well as the torsional stress controlled by the length of the spacer region[18,19,38–41]. Bound RNAP/σ[70] then undergoes a conformational change that catalyzes double-stranded DNA separation, creating a transcription bubble that initially encompasses half the −10 motif, the Disc, and the first two nucleotides of the ITR[39,40]. RNA polymerization begins at a TSS determined by where the catalytic site in the β subunit contacts the DNA template, canonically at position +1. The transcription bubble is then stabilized by interactions in the Disc[22] and the formation of an R-loop, whereby the newly synthesized RNA strand immediately hybridizes to the DNA template[42]. The ITR sequence controls the R-loop's thermodynamic stability. Finally, transcription initiation is successful once enough DNA is pulled into the stable transcription bubble that the accumulated stress exceeds the interaction strength between RNAP/σ[70] and promoter DNA, causing promoter escape and a transition to processive RNA synthesis[25,42,43].

From these interactions, we formulated a statistical thermodynamic model of transcriptional initiation that accounts for competitive binding of RNAP/$\sigma^{70}$ to all DNA and the multiple sequence contacts that control RNAP/$\sigma^{70}$ recruitment at each promoter (**Methods**). So long as the internal states do not become abundant, for example, by significant transcriptional pausing[44], the model indicates that we can decompose how a promoter's sequence controls the interaction energies into a sum of free energies that can be related to the transcription initiation rate (*TX*), according to:

$$\Delta G_{total} = \Delta G_{UP} + \Delta G_{-35} + \Delta G_{spacer} + \Delta G_{-10ext} + \Delta G_{-10} + \Delta G_{disc} + \Delta G_{ITR} \tag{1}$$

$$\log\left(\frac{TX}{TX_{ref}}\right) = -\beta(\Delta G_{total} - \Delta G_{total,ref}) \tag{2}$$

where $\Delta G_{total}$ is the difference in free energy between an unbound promoter and a promoter- RNAP/$\sigma^{70}$ complex with a stable transcriptional bubble, which is used to predict a promoter's TX rate in comparison to a reference promoter sequence with calculated $\Delta G_{total,ref}$ and measured $TX_{ref}$. $\beta$ is a measurable model constant that converts free energies into state probabilities. Here, we arbitrarily set $\Delta G_{total,ref}$ to zero so that stronger and more favorable interaction free energies have more negative values as compared to the reference promoter, which has a low transcription rate.

We designed the 14,206 promoter sequences to measure how each motif sequence alters these free energies to create a sequence-complete model. The baseline promoter sequence contained consensus hexamers, consensus spacer length, and an optimized background sequence generated to match the G/C content of the *E. coli* genome, while eliminating cryptic hexamer motifs and restriction cut sites. The changes to the baseline promoter sequence included all possible −10 hexamer motifs ($4^6 = 4096$ variants), all possible −35 hexamer motifs ($4^6 = 4096$ variants), all possible −10 extended motifs placed within 8 combinations of consensus and anti-consensus hexamers ($4^4 \times 8 = 2048$ variants), 229 spacer sequences with varied lengths and nucleotide compositions, 605 UP sequences with varied AT content placed within 4 combinations of consensus and anti-consensus hexamers ($605 \times 4 = 2420$ variants), 735 discriminator variants with varied lengths and GC content, and 582 ITR variants with varied R-loop stabilities and pause elements (Fig. 1B). Using oligopool synthesis and two-step library cloning, we constructed a barcoded plasmid pool that uses each promoter in a common genetic context, expressing a single protein with a moderate translation initiation rate (about 5000 on the RBS Calculator v2.1 scale[5]). Barcodes were designed to have pair-wise Hamming distances of 2 or greater. To avoid potentially confounding effects, we used the two-step cloning procedure to position the barcodes within the 3′ untranslated region.

We then carried out in vitro transcription reactions, TSS mapping, and next-generation sequencing to measure the TX rate of each promoter at each TSS, utilizing a minimal system containing only RNAP/$\sigma^{70}$, the barcoded plasmid pool, NTPs, and buffer (Methods). TSS mapping was performed by harvesting product RNA, ligating a 5′ RNA adapter, converting to cDNA, circularizing, using PCR to generate amplicons containing the barcode and TSS, and obtaining over 323 million barcode-TSS mapped reads from Illumina sequencing. Across all promoters, we found 182 TSSs with at least 1000 mapped reads, revealing a primary TSS region and a less frequent, secondary TSS region arising from a downstream cryptic promoter. TX rates were then measured by carrying out DNA-Seq and RNA-Seq on triplicate in vitro transcription reactions, separately harvesting DNA and RNA, converting RNA to cDNA, using PCR to generate barcode-containing amplicons, and obtaining between 81.3 and 130.5 million barcoded reads per reaction. From the RNA/DNA read count ratios, we obtained

TX rates for 13480 promoter variants (mean and standard deviation), excluding any with fewer than 50 read counts in any replicate. Replicate read count measurements were highly reproducible ($R^2 = 0.89–0.99$, Supplementary Fig. 1) with 5391 high-precision TX rates (coefficient of variation < 0.40). All promoter sequences, DNA read counts, RNA read counts, TX rates, and TSS frequencies are found in the Supplementary Data 1.

## Model training and validation

We began model training by identifying 5193 promoter variants where RNAP/$\sigma^{70}$ bound to a single site with one predominant TSS, enabling us to unambiguously pinpoint their motif sequences (Fig. 1C and Supplementary Fig. 2). The TX rates for these single-site promoters altogether varied by 123-fold with sizable effects from each individual motif (Fig. 1D). Importantly, endogenous RNases were not present in the in vitro transcription reactions, enabling us to vary the Disc and ITR sequences without changing the mRNA's stability, which is not possible in equivalent in vivo measurements. We started the model development by specifying 472 sequence, structural, and energetic properties that relate how each motif sequence contributes to each free energy term (Supplementary Table 1). For example, for all UP sequences, we calculate the minor groove width of the distal and proximal UP sites[45]; for all ITR sequences, we calculate the thermodynamic stability of the R-loop[46,47]; for all spacer sequences, we calculate the local DNA rigidity[48]. As categorical properties, we split the −35, −10, and Disc motifs into six 3-nucleotide regions and include 384 3-mers. We split the −10 ext motif into 2-nucleotide regions and include 32 2-mers. We also include the spacer length as a categorical property.

We then randomly split our dataset into a training set (4673 promoters, 90%), carried out tenfold cross-validation to identify the optimal hyperparameters of a machine learning model, and tested its accuracy on the remaining unseen test set (520 promoters, 10%). We evaluated several models (Supplementary Table 2) that use regularization to identify the optimal coefficients of a linear, additive model that mirrors our free energy parameterization (Eq. 1) with dataset normalization that mirrors our statistical thermodynamic model (Eq. 2). We carried out feature reduction to remove extraneous properties, which lowered the number of fitted coefficients. The pruned properties included the first 2-nucleotide region of the −10ext motif and the last 3-nucleotides of the Disc motif, which had no discernable effect on TX rate in this dataset.

Overall, we found that a ridge regression model with 346 fitted coefficients yielded a convergent learning curve (Fig. 2A) and highly accurate predictions ($R^2 = 0.80$, Fig. 2B) with similarly low error distributions (Fig. 2D) across both the training and unseen test datasets. We then performed ANOVA analysis to quantify how each predicted free energy term contributed to the promoters' measured TX rates in our dataset and found that 83% of the TX rate variance could be explained by varying the interactions that control initial RNAP/$\sigma^{70}$ recruitment, using the UP, −35, −10, −10 ext, and spacer motifs (Fig. 2C). In contrast, 7.1% of the TX rate variance was explained by differences in the interactions controlling DNA melting, R-loop formation, and promoter escape, which are affected by the Disc and ITR regions.

A key benefit of our hybrid biophysical-machine learning approach is that the fitted coefficients quantify the physical interactions between RNAP/$\sigma^{70}$ and each promoter motif (Fig. 2E), including all motif sequences and enabling direct comparisons to already known canonical interactions as positive controls. For example, without human supervision, our approach correctly identified the canonical −35 motif (TTGACA), −10 motif (TATAAT), extended −10 motif (TG), and the optimal spacer length (17 base pairs). We also found that AT-rich distal and proximal UP sites had more favorable interactions and that GC-rich Disc motifs were enriched, which were previously

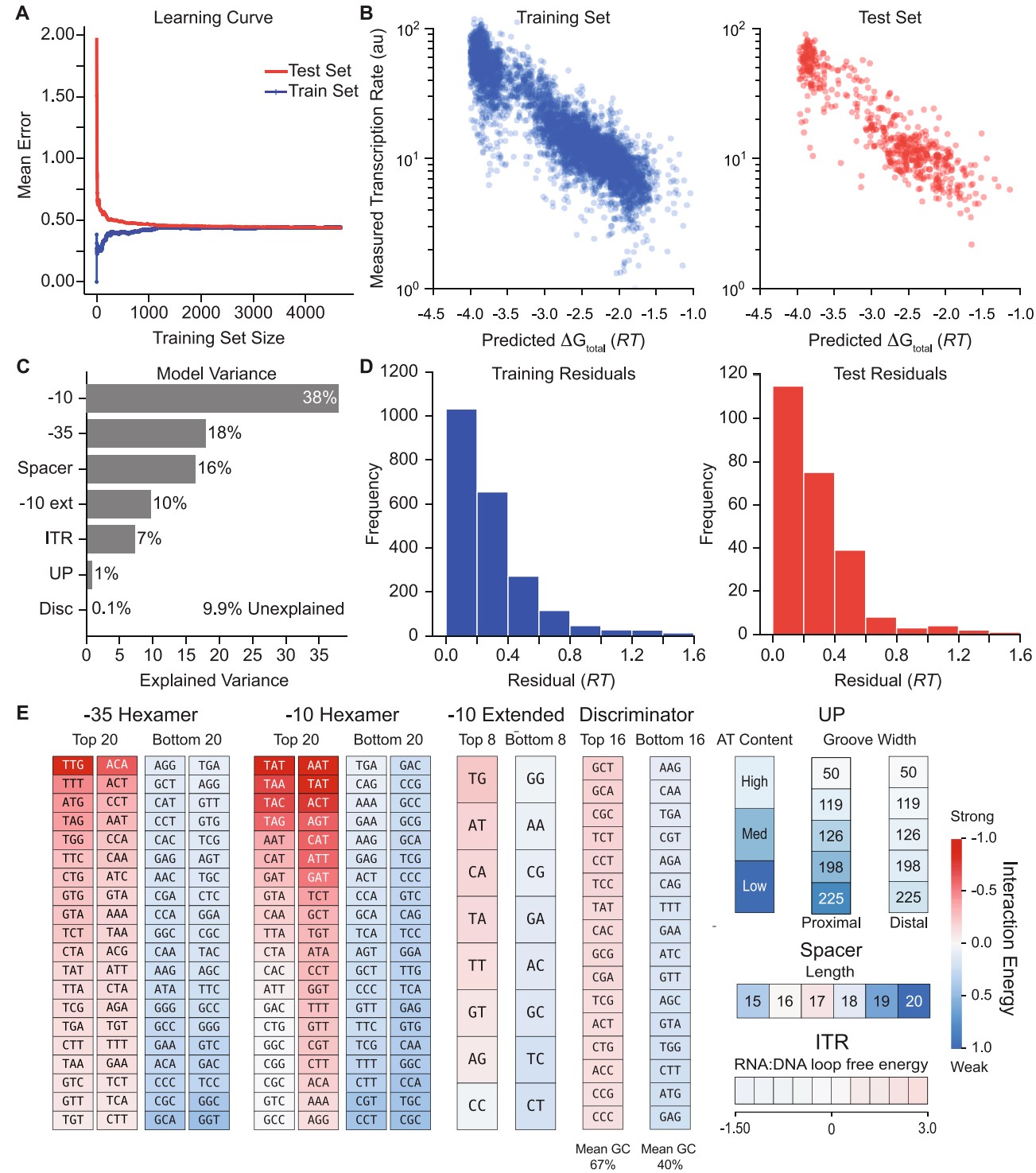

**Fig. 2 | Model Development and Validation using Machine Learning. A** A learning curve shows the training and testing of a ridge regression model to identify the unknown interaction energies. **B** Model-predicted free energies are compared to measured transcription rates for both (left) the training set and (right) the unseen test set. Pearson correlation coefficient ($R^2$) is equal to 0.80 for both the training and test sets. **C** The explained variances for each promoter interaction are

shown. **D** Histograms of model error are shown for the training and test sets, using energy units. Mean absolute error (MAE) is equal to 0.27 $RT$ for the training set (left) and 0.28 $RT$ for the test set (right). **E** The learned interaction energies are shown for the strongest and weakest ones. A poster-sized schematic showing all interaction energies is available (Supplementary Information). Data are provided in Supplementary Data 1.

observed[18,22]. However, most promoters do not contain canonical motif sequences; their TX rates are controlled by a mixture of weaker interactions. A key aspect of our model is its complete set of interaction energies between RNAP/$\sigma^{70}$ and double-stranded DNA, covering all possible promoter sequences, which provides the ability to predict the TX rate of any $\sigma^{70}$ promoter. All interaction energies are available in

Supplementary Data 1, including numerical values and a poster-sized chart.

## Model generality

We then tested the model's generalizability and accuracy across 22,132 characterized promoters, using sequence information alone to predict

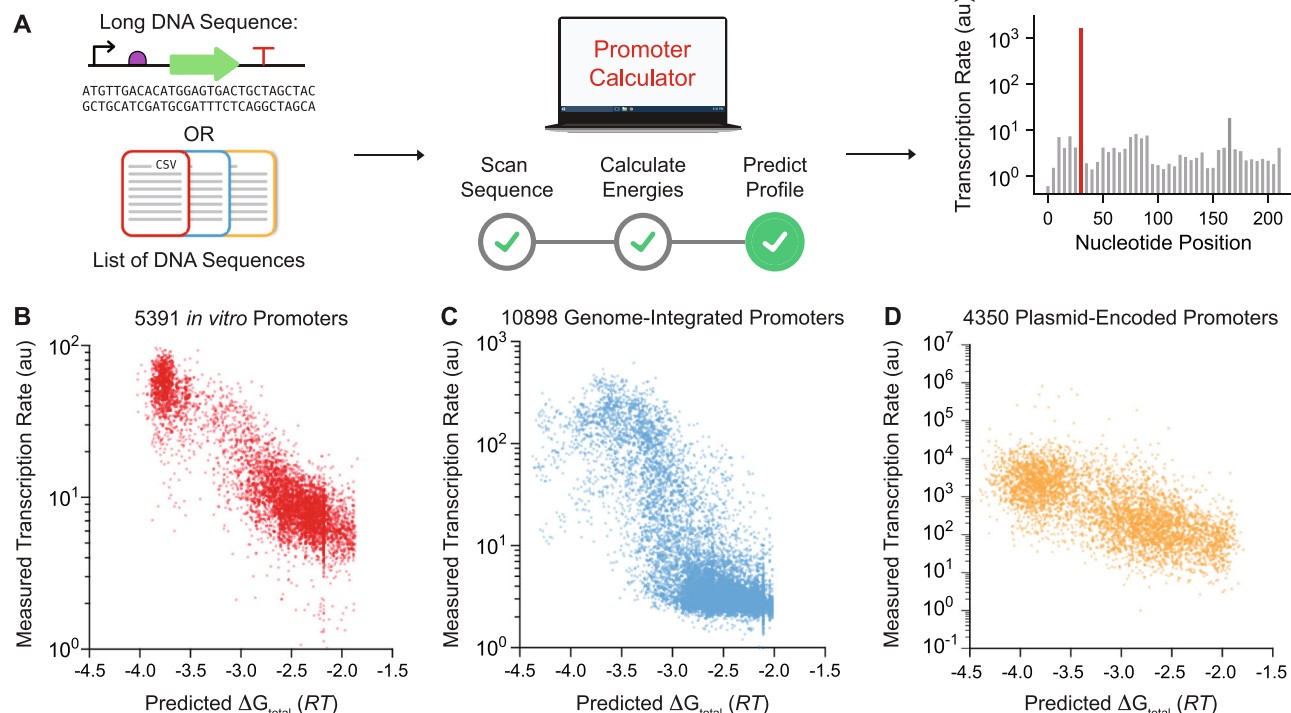

**Fig. 3 | Validation of Sequence-to-Function Model on Diverse Promoters.**
**A** Arbitrary DNA sequences are inputted into the model to predict its transcriptional profile (transcription rates vs. nucleotide positions) without start site information. **B** Model comparisons on 5391 designed promoters (LaFleur et al., this study) are compared to in vitro transcription rate measurements ($R^2 = 0.79$, Spearman's $\rho = 0.80$, MAE = 0.33 $RT$, MSE = 0.18 $RT$). **C** Model predictions on 10898 genome-integrated modular promoters[16] characterized by Urtecho et al. are compared to in vivo transcription rate measurements ($R^2 = 0.60$, Spearman's $\rho = 0.67$, MAE = 0.93 $RT$, MSE = 1.28 $RT$). **D** Model predictions on 4350 non-repetitive plasmid-encoded promoters[12] characterized by Hossain et al. are compared to in vivo transcription rate measurements ($R^2 = 0.45$, Spearman's $\rho = 0.69$, MAE = 1.08 $RT$, MSE = 1.88 $RT$). MAE and MSE were determined by fitting a proportionality constant (best-fit slope) accounting for experimental variation. Data are provided in Supplementary Data 1.

their transcriptional profiles, specifically their TX rates for each potential TSS position (Fig. 3A). Here, we do not input the promoters' actual TSS positions and motifs, and instead combine the interaction energies with statistical thermodynamics to identify the most likely RNAP/$\sigma^{70}$ binding site configuration. For each potential TSS position, we scan the surrounding DNA sequence and enumerate the several ways that RNAP/$\sigma^{70}$ may bind to it, varying the spacer and Disc lengths with corresponding changes in motif sequences. We apply the model's interaction energies to calculate the $\Delta G_{total}$ for each configuration (Eq. 1) and determine the configuration with the most negative $\Delta G_{total}$. We repeat these calculations for each potential TSS position within the inputted DNA sequence.

We carried out these additional tests on four datasets: (1) 5391 designed promoters with high-precision TX rate and TSS measurements, characterized here using in vitro RNAP/$\sigma^{70}$ transcription assays (LaFleur et al.); (2) 10898 promoters using combinations of motifs to express a ribozyme-insulated transcript, characterized using genome-integrated test circuits inside *E. coli* cells (Urtecho et al.)[16]; (3) 4350 non-repetitive promoters using highly diverse sequences to express a transcript, characterized using test circuits carried on multi-copy plasmids inside *E. coli* cells (Hossain et al.)[12]; and (4) 1493 inducible promoters that contain transcription factor binding sites and express a ribozyme-insulated transcript, also characterized using genome-integrated test circuits inside *E. coli* cells (Yu et al.)[17]. We first confirmed that the statistical thermodynamic model accurately predicted free energies of the in vitro data collected by LaFleur et al. without utilizing TSS measurements ($R^2 = 0.79$, Spearman's $\rho = 0.80$, Fig. 3B). Next, we tested the model's predictions on the three in vivo datasets, which were not used during model development. We found that the statistical thermodynamic model retained accuracy ($R^2 = 0.60$, Spearman's $\rho = 0.67$; $R^2 = 0.45$, Spearman's $\rho = 0.68$; $R^2 = 0.65$, Spearman's

$\rho = 0.70$; Fig. 3C, D and Supplementary Fig. 3), even though in vivo mRNA levels are affected by other potentially confounding factors, for example, variation in DNA copy number, alternate/multiple sigma factor binding sites, and mRNA stability. Lastly, we evaluated the model's accuracy on each dataset by calculating the MAE across 20 evenly spaced transcription rate bins (Supplementary Fig. 4). We found the model maintains high accuracy across the broad range of transcription rates tested and that rate-binned MAEs were primarily dependent on the number of promoters within each bin.

To evaluate the model's predictions using additional types of measurements, we then selected ten of the non-repetitive promoters collected by Hossain et al. and characterized their expression levels in test genetic circuits inside *E. coli* cells, utilizing RT-qPCR to measure their mRNA levels and flow cytometry to measure their fluorescent reporter protein levels. We found that the model's predictions for these promoters were highly proportional to their measured mRNA levels ($R^2 = 0.75$, Spearman's $\rho = 0.87$) and fluorescent protein expression levels ($R^2 = 0.78$, Spearman's $\rho = 0.87$) over a 78640-fold dynamic range (Supplementary Fig. 5). All dataset sequences, model predictions, and TX rate measurements are included in Supplementary Data 1.

## A non-linear interaction model to test for inter-motif cooperativity

Next, we developed a non-linear free energy model to test whether the incorporation of inter-motif interactions could improve the accuracy of transcription rate predictions. To do this, we utilized the 10,898 promoters characterized by Urtecho et al. as our training dataset. We considered all possible pair-wise interactions (quadratic terms) that can occur between the six promoter regions, which added 30 unknown coefficients. We then applied linear regression to determine the values

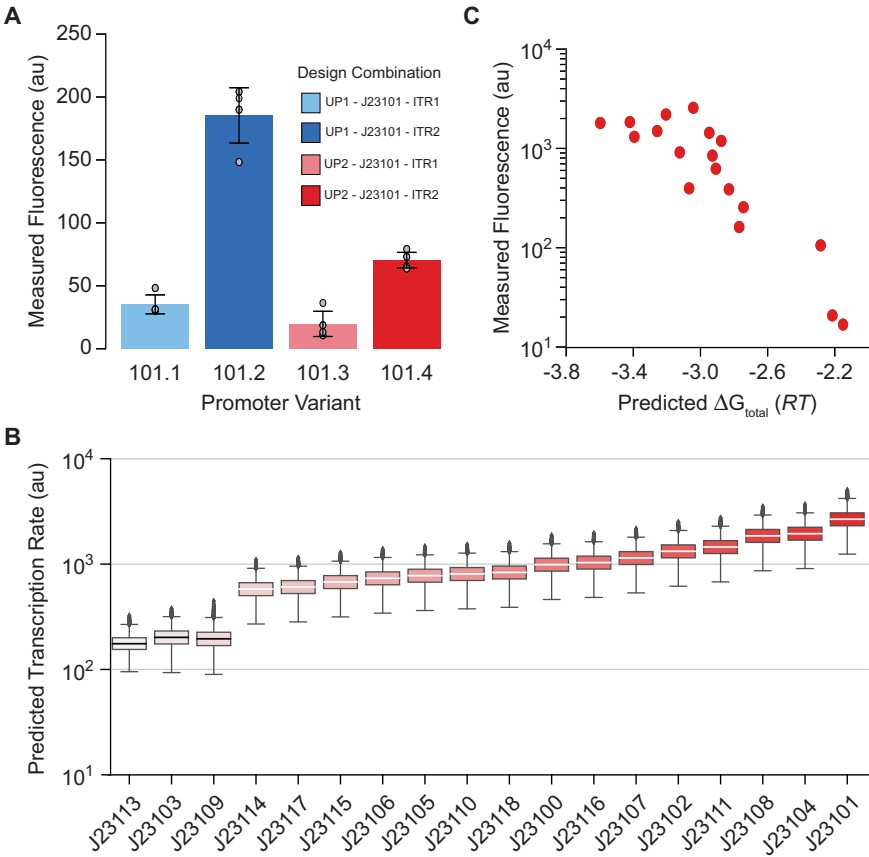

**Fig. 4 | Promoter Context Effects. A** Measured in vivo transcription rates for the J23101 promoter genetic part with varied upstream (UP) and downstream (ITR) sequences, showing transcriptional context effects. Gray dots denote independent measurements ($n = 4$ biological replicates). Bars denote each variants measured mean ± standard deviation. **B** Transcriptional context effects are quantified for common promoter genetic parts, showing predicted transcription rates when varying 30 bp of the upstream and downstream sequences. Distributions were created using 10,000 simulations ($n = 10,000$). Box bounds were defined by the first and third quartiles of the distribution, center lines by the median, whiskers by the minima and maxima, and black dots by outliers. **C** Model predictions are compared to in vivo fluorescence measurements for these promoters when specifying their upstream and downstream sequences ($R^2 = 0.79$, Spearman's $\rho = 0.85$). Data are provided in Supplementary Data 1.

of these coefficients, followed by model testing. When measuring accuracy using Pearson $R^2$, we found that the non-linear model had higher accuracy on the Urtecho training dataset ($R^2 = 0.69$ vs. $R^2 = 0.60$), but had similar accuracies when tested on the LaFleur and Hossain test datasets ($R^2 = 0.76$, $R^2 = 0.46$) as compared to the linear free energy model ($R^2 = 0.79$, $R^2 = 0.45$). When measuring accuracy using MAE or MSE, the non-linear model had lower accuracies on the LaFleur and Hossain test datasets (Supplementary Table 4). Altogether, the addition of the pair-wise quadratic terms to the free energy model did not improve model accuracy, which suggests the absence of quadratic cooperative or anti-cooperative interactions between motifs within the same promoter region. For subsequent results, we continue to utilize the linear free energy model.

**Promoter context effects**

Across microbial biotechnology, promoters are routinely treated as reusable, swappable genetic parts to control protein expression levels[6]. However, the model predicts that the DNA sequences surrounding a promoter will greatly affect its TX rate, specifically the UP and ITR regions, which would explain why some promoters express certain proteins at much lower levels. We next tested the model's ability to predict, explain, and overcome these genetic context effects. First, we designed UP and ITR sequences that the model predicted would greatly alter a promoter's TX rate, inserted them around the commonly used J23101 promoter, and measured their in vivo TX rates using a fluorescent protein reporter inside *E. coli* cells (Methods).

Changing only the UP region or only the ITR region reduced the promoter's TX rate by up to 5.3-fold, while changing both regions reduced the promoter's TX rate by up to 9.4-fold (Fig. 4A). We next evaluated whether these genetic context effects are potentially widespread by carrying out Monte Carlo simulations to predict the range of TX rates expected when varying 30 bp of the UP and ITR regions of several commonly used promoters (Fig. 4B). We found that all promoters exhibited sensitivity to genetic context with an overall average coefficient of variation of 0.21. We then tested whether the model is able to correctly account for genetic context effects and predict the promoters' activity in a particular context. To do this, we selected a classic community dataset whereby the in vivo TX rates of these commonly used promoters were previously characterized inside *E. coli* cells and within a specific genetic context[4]. We then inputted the specific UP, core promoter, and ITR sequences from this dataset into our model and confirmed accurate predictions ($R^2 = 0.79$, Spearman's $\rho = 0.85$, Fig. 4C), further illustrating the model's generality.

**Forward engineering promoters**

Next, we developed an automated optimization algorithm that designs promoter sequences with desired transcriptional profiles, taking into account both the upstream DNA sequence and the transcribed mRNA sequence (Fig. 5A). The algorithm identifies optimal promoter sequences that provide user-defined TX rates at desired TSSs, while minimizing the TX rate at undesired, off-target TSSs (Methods). We tested this algorithm by designing, constructing, and characterizing

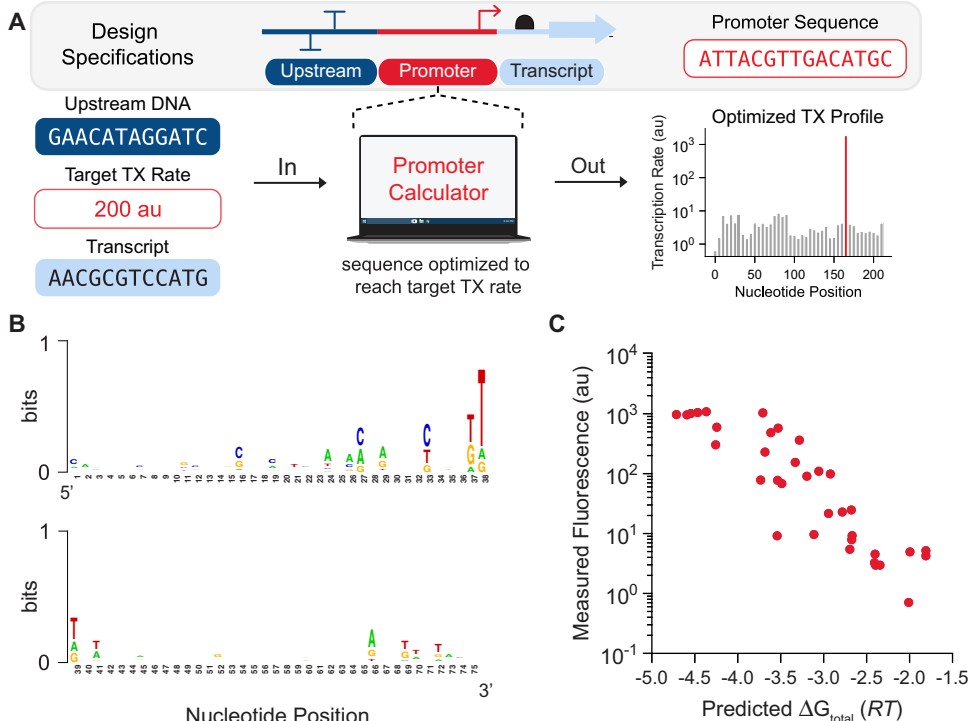

**Fig. 5 | Automated Design of Synthetic Promoters. A** Promoter sequences were forward engineered with desired transcription rates and start sites. **B** Sequence analysis of the 75 bp forward engineered promoters showed low sequence similarity. **C** Model predictions are compared to in vivo transcription rate measurements for the 35 designed promoters ($R^2 = 0.80$, Spearman's $\rho = 0.89$). Red dots denote the mean of duplicate measurements ($n = 2$ biological replicates). Data are provided in Supplementary Data 1.

35 synthetic promoter sequences with targeted, systematically varied TX rates. The designed promoters were 60–77 base pairs long and shared only ~27% sequence similarity (Fig. 5B and Supplementary Fig. 6). The model predicted single predominant TSSs at the desired locations, though it becomes progressively more difficult to achieve single-TSS predominance as the targeted TX rate is lowered. We then measured the promoters' TX rates, each expressing a ribozyme-insulated transcript on a multi-copy plasmid inside *E. coli* cells during exponential growth (Methods), and found that the model's predictions were highly accurate ($R^2 = 0.80$, Spearman's $\rho = 0.89$, Fig. 5C), enabling fine control of site-specific transcription rates over a 1525-fold range, while taking into account the surrounding genetic context.

**Genetic circuit debugging and RNAP flux analysis**

Finally, we demonstrated how the statistical thermodynamic model facilitates the debugging of large genetic circuits by predicting their transcriptional profiles and identifying cryptic promoters that could disrupt the circuit's function. As a demonstration, we selected a recently characterized genetic circuit that uses 11 engineered promoters and 7 transcription factors to carry out digital logic[8] (Fig. 6A). The model predicted single-TSS peaks for 9 of the engineered promoters and identified 51 cryptic promoters (18 sense and 33 antisense). These predictions were qualitatively confirmed by system-wide RNA-Seq measurements and transcriptional flux inferences with an overall accuracy of 55% for cryptic promoter identification (Methods). To further demonstrate the utility of the model, we compared model predicted transcriptional profiles to the measured RNAP flux on a portion of the circuit that showed leaky expression in the OFF state. Across this 1000 bp region using the $P_{BAD}$ promoter to express AmtR, the model correctly predicted the presence of cryptic TSSs with high transcription rates that led to higher than expected mRNA levels (Fig. 6B). By combining the transcription rate predictions with automated computational optimization, we then redesigned the AmtR coding sequence, varying its synonymous codon usage to minimize its transcriptional profile, which removed the cryptic promoters (Fig. 6C). We then demonstrated that our automated algorithm can also generate no-promoter regions with minimized transcription rates by designing 60 bp and 120 bp regions and measuring their transcription rates using a reporter protein in test genetic circuits (Methods). We found that both no-promoter regions yielded very low reporter expression levels (only 7.6% above white cell autofluorescence and 161-fold lower reporter expression than the commonly used J23100 promoter when corrected for white cell autofluorescence) (Fig. 6C inset).

## Discussion

We developed a biophysical model of RNAP/$\sigma^{70}$-promoter interactions −the Promoter Calculator− to predict site-specific transcription initiation rates across any $\sigma^{70}$ promoter sequence. We first trained and tested the model by carrying out in vitro transcription rate measurements and TSS mapping on thousands of designed promoter sequences, followed by additional validation on thousands of promoters characterized inside cells. Overall, the model contains 346 transparent parameters to calculate the strengths of the interactions at the −10 hexamer, the −10 extended motif, the −35 hexamer, the upstream element (UP) element, the discriminator, the spacer region, and the ITR with validated predictions across 22,132 diverse promoter sequences. We then applied the model to design synthetic promoter sequences with targeted transcription initiation rates and to debug sources of cryptic transcription in engineered genetic systems. The model provides a more complete understanding of how both canonical and non-canonical motifs collectively control transcription rates across all potential DNA sites, yielding the transcriptional profile of a genetic system.

Careful experimental design was needed to maximize the information content of our transcription rate measurements. First, by carrying out in vitro transcription assays using only RNAP/$\sigma^{70}$ enzyme, we

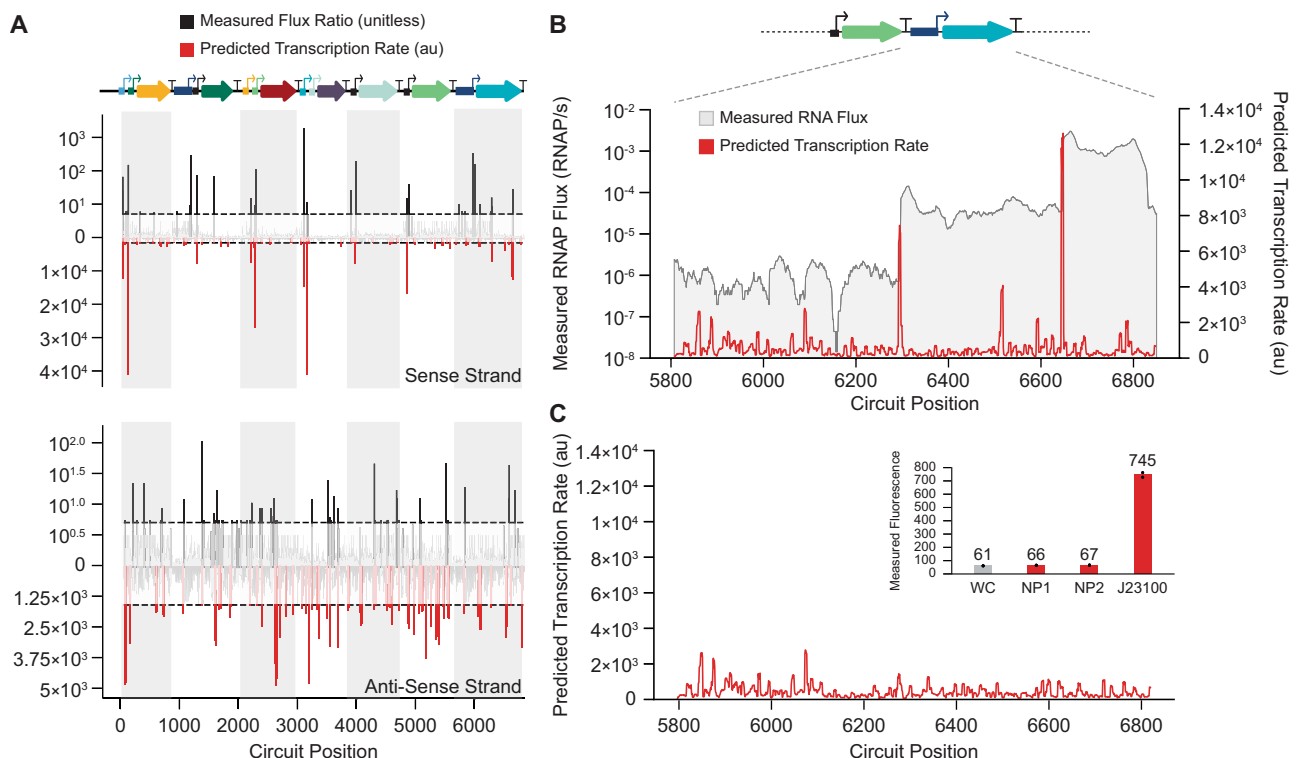

**Fig. 6 | Genetic Circuit Debugging Through Cryptic Promoter Identification. A** The predicted σ⁷⁰ transcriptional profile of an 11-promoter genetic circuit[8] is compared to in vivo transcription rate and start site measurements on the (top) sense and (bottom) anti-sense strand. Transcription flux ratios are the measured differences in adjacent mRNA levels from RNA-Seq. White and gray shadows correspond to each transcribed cistron. The horizontal dotted black lines show the minimum transcription rates that define a start site. Experimental TSS cutoffs are previously described[8], and prediction cutoffs are defined in the Methods. Annotated start sites are depicted in the circuit diagram with arrows. **B** The predicted σ⁷⁰

transcriptional profile for the P_BAD-amtR portion of the circuit in the OFF state. Red lines are model predictions, and gray overlays show measured RNAP flux. **C** The predicted σ⁷⁰ transcriptional profile for a redesigned amtR protein coding sequence that minimized the transcription initiation rate inside the coding sequence. (inset) Measured fluorescence levels for designed no-promoter regions predicted to have minimal transcription rates (NP1: 120-bp, NP2: 60-bp) as compared to (WC) white cells and the J23100 promoter. Black dots denote independent measurements. Bars denote the mean across duplicate measurements (*n* = 2 biological replicates). Data are provided in Supplementary Data 1.

eliminated several competing interactions that routinely confound in vivo transcription rate measurements. For example, our assays did not contain alternative σ-factors or RNases, resulting in mRNA level measurements that were solely controlled by RNAP/σ⁷⁰ interactions during transcription. Second, by employing oligopool synthesis, we were able to rationally design the thousands of promoter sequences needed to perturb individual interactions and measure their effects on transcription rates. Third, we designed all barcodes to have dissimilar sequences (pair-wise Hamming distances 2 or greater) to avoid the possibility of mis-mapping barcodes during data analysis, which would lead to non-random biases in the transcription rate measurements. Finally, the transcription start site mapping was found to be essential to pinpointing sequence motifs for model training, particularly for weaker promoters that often exhibited multiple transcription start sites. Altogether, our experimental design, carried out in biological triplicate, yielded a firm ground truth on which to build and test the predictive model. Notably, our in vitro experimental approach can be readily extended to other sigma factors and organisms to measure the interaction strengths between their respective RNAP/σ complexes and double-stranded DNA. However, our in vitro transcription assays did not contain transcription factors and did not measure regulated transcription, although it is feasible to add purified transcription factors to study their regulatory effects.

A key advantage to our hybrid modeling approach, in contrast to non-interpretable machine learning, was the extraction of coefficient values that directly correspond to the strengths of the RNAP/σ⁷⁰-DNA interactions (Fig. 2, Supplementary Poster). In fact, using only these interaction free energies in a numerical table (Supplementary Data 1),

it is straightforward for researchers to predict transcription rates and design synthetic promoters without the use of a computational framework. Furthermore, the interaction free energies are all reported on the same scale, enabling one to rank the importance of each motif toward controlling transcription initiation rates.

We also tested our model accuracy on several in vivo datasets. Overall, we found higher model accuracy when the in vivo dataset's experimental design yielded more precise transcription rate measurements. For example, Urtecho et al. integrated all test genetic circuits into the *E. coli* genome to reduce the variation in DNA copy number. They also utilized self-cleaving ribozymes to reduce the effects of RNase activity, though even the most active ribozymes have efficiencies of about 90%[35–37]. In contrast, Hossain et al. encoded all test genetic circuits on multi-copy plasmids and did not utilize a ribozyme, leading to significant stochastic cell-to-cell variation in mRNA levels. Both in vivo datasets were affected by the presence of alternative σ-factors, which can potentially initiate transcription at motifs outside the model parameterization. Correspondingly, model accuracy was highest on the in vitro dataset (LaFleur, $R^2$ = 0.79), followed by the in vivo datasets (Urtecho, $R^2$ = 0.60; Hossain, $R^2$ = 0.45).

However, we found that controlling for confounding factors in the Urtecho and Hossain in vivo datasets led to even higher model accuracies. For example, changes to the promoter ITR can impact translation rates, which then indirectly affect mRNA levels by altering ribosome protection and the mRNA transcripts' decay rates[28,49,50]. To assess this effect, we applied the RBS Calculator[5] to predict the translation initiation rates of all mRNAs and then analyzed model accuracy on systems that exhibited high translation rates where mRNA

decay contributes the least. We found that model accuracy improved in both cases (Urtecho, $R^2 = 0.61$; Hossain, $R^2 = 0.50$, Supplementary Fig. 7). The opposite effect was observed when mRNAs were predicted to have low translation rates; model accuracies were reduced (Urtecho, $R^2 = 0.51$; Hossain, $R^2 = 0.43$, Supplementary Fig. 7). These observations suggest that combining multiple models together, predicting transcription, translation, and mRNA decay rates, is a viable route to improving accuracy.

Surprisingly, in our set of forward engineered promoters characterized in vivo, we found five examples of very strong promoters that may have reached a plateau in maximal transcription rate where the recruitment of RNAP/$\sigma^{70}$ and/or formation of a stable open complex is no longer the rate-limiting step (Fig. 5C). These promoters have predicted free energies of −4.4 $RT$ or stronger and further strengthening of these interactions only resulted in the same reporter expression levels. Instead, it is possible that another downstream step in transcription becomes rate-limiting, for example, the transition from a stable open complex to an active elongation complex. However, we do not observe a reduction in reporter expression levels using such strong promoters, which could occur if overly strong interactions inhibited RNAP's ability to escape the promoter region.

During our model development and training, we also identified several alternate models that were nearly as accurate as the final one that we selected. For example, instead of splitting the −10 and −35 hexamers into four 3-mers (Fig. 2), we also developed free energy models that utilized position weight matrices, where the RNAP/$\sigma^{70}$ interactions with each single position in the hexamers were quantified using 4 coefficients (one per A, C, G, T base pair). However, this mono-nt model had lower accuracies on the LaFleur in vitro dataset as well as the Urtecho and Hossain in vivo datasets (Supplementary Data 4). To explain why, we compared the models' interaction free energies and found that the 3-mer model identified favorable interactions within some non-canonical 3-mers located in the upstream region of the −10 hexamer (Supplementary Fig. 8). Separately, we also developed and tested a non-linear free energy model that included all possible pairwise inter-motif interactions (Methods). However, even with the addition of these quadratic terms, the non-linear model's accuracy was only higher on its training set (Urtecho, $R^2 = 0.69$) and was similar or lower on other datasets (LaFleur, $R^2 = 0.76$; Hossain, $R^2 = 0.46$; Supplementary Fig. 9; Supplementary Table 4). These results suggest that it will be necessary to design and characterize new datasets to comprehensively test for the presence of cooperative inter-motif interactions within the same promoter region.

We next benchmarked our model against a recently developed model of transcription initiation[15] that was trained using Flow-Seq (Sort-Seq) characterization of promoters with randomly dispersed mutations or random sequences. This model calculates the proportion of time RNAP/$\sigma$ spends bound to a promoter sequence using an energy matrix, which is then related to the measured transcription rates using several fitted parameters, including an apparent chemical potential, an effective slope, an effective intercept, and an optimized detection threshold. Overall, when carrying out equivalent data analysis (Methods), our model achieves higher accuracy on the LaFleur, Urtecho, and Hossain datasets (Supplementary Table 4). We also demonstrated our model's generality by designing synthetic promoters with highly dissimilar sequences and targeted transcription rates with high accuracy (Fig. 5C). All benchmark model calculations, measurements and F-statistics are included in Supplementary Data 1, 2, 3, and 4.

Further model improvements will require taking into account additional long-distance and non-additive transcriptional interactions, for example, interference between colliding RNAPs[51–53] and sequence-dependent sensitivities to DNA supercoiling[54,55], as well as additional gene expression processes that ultimately affect measurements, such as changes in DNA copy numbers, transcriptional regulation, translation rates, mRNA decay rates, and coupling between these processes[5,28,56,57]. A key necessity will be to ensure both model accuracy and model generality by evaluating predictions across many datasets covering the full range of potential interactions. Overall, the best approach to developing future model improvements will be to design and characterize promoter datasets that specifically perturb insufficiently characterized interactions to measure their strengths.

In conclusion, we developed a parsimonious (human-understandable) biophysical model of bacterial transcription initiation with demonstrated accuracy across thousands of $\sigma^{70}$ promoters with highly diverse sequences, enabling site-specific control over transcription initiation rates in engineered genetic systems. Our model shows how multiple weak interactions contribute to RNAP/$\sigma^{70}$ transcriptional control, including its promiscuous activity on DNA sequences without cognate binding motifs[58]. Our bottom-up model-building approach is readily extendable to other RNAP/$\sigma$ complexes and demonstrates how advances in machine learning can enhance, rather than replace, existing thermodynamic formalisms with the overall goal of creating a universal system-wide language for engineering gene regulation in synthetic genetic systems.

## Methods

### Library design
14206 $\sigma^{70}$ promoters were rationally designed to include 2420 UP elements, 4096 −35 hexamers, 229 spacers, 2048 −10 extended motifs, 4096 −10 hexamers, 735 discriminators, and 582 ITRs. The 2420 UP element sequences were designed in two subsets: (i) the first set varied adenine and thymine (AT) content from 0 to 100% within a background sequence containing combinations of consensus, anti-consensus, and mutated −10 and −35 hexamers; and (ii) the second set introduced short cytosine and adenine repeats into the distal and proximal binding sites, while varying AT content within a background sequence containing the same combinations of consensus, anti-consensus, and mutated hexamers. The −10 hexamer set was designed to include the 4096 6 nt sequences in the −10 hexamer, a consensus −35 hexamer, and a 17-bp spacer. The −35 hexamer set was designed to include the 4096 possible 6 nt sequences in the −35 hexamer, a consensus −10 hexamer, and 17 bp spacer. The 229 spacers were designed by varying the spacer length from 1- to 32 bp, in between a consensus −35 and a consensus −10 hexamer, while varying the spacer nucleotide composition at each length. The −10 extended promoter variants were designed to include the 256 possible 4 nt sequence at positions −14 to −17 within 8 different background sequences containing combinations of consensus, anti-consensus, and mutated −10 and −35 hexamer sequences. The 735 discriminator variants were designed in two sets with a background sequence containing a consensus −10 hexamer: (i) a first set that varied the guanine-cytosine (GC) content from 0 to 100% and the discriminator length from 6 to 8 bp with 5 randomly generated sequences satisfying each criterion; and (ii) a second set that introduced all possible 3 nt sequences in the first half of the discriminator and varied the GC content in the remaining 4 nt. The 582 ITR variants were designed to vary the number of purine bases, the GC content, and the presence of −10 hexamer-like pause sequences within the ITR. Unless otherwise stated, promoter variants were designed within a background containing consensus hexamers and a canonical 17 bp spacer. A complete list of the promoter sequences is available in the Supplementary Data 1.

### Oligo pool design and optimization
An oligopool containing a mixture of 170 nt oligonucleotides was synthesized (Genscript). Each oligonucleotide design contained a promoter variant, four restriction enzyme cut sites, two primer binding sites, and a unique 20-nucleotide barcode sequence (Supplementary Fig. 10). A custom design algorithm was used the create barcodes and primer binding sites. Barcodes were designed to maximize pairwise hamming distance between each other and with respect to the

unique *k*-mer list generated from the 14206 designed promoters and the *E. coli* genome. Primer binding site sequences were designed to have target melting temperatures of 55 °C, low probabilities of forming primer dimers, and minimal primer structure. Primer binding site sequences were also designed with maximum dissimilarity to barcode sequences, promoter sequences, and the *E. coli* genome sequence.

## Library cloning and growth

PCR was carried out using the oligopool as DNA template to create an insertion cassette library, using Q5 High-Fidelity DNA Polymerase (NEB), 20 cycles, and primers 1 and 2 (Supplementary Table 3). DNA was extracted and purified via gel electrophoresis. The purified insertion cassette library and plasmid pFTV1 were double-digested using BamHI and HindIII (NEB), purified, ligated, purified again, and then transformed into NEB 5-alpha Electrocompetent *E. coli* (NEB). All ligations used T7 DNA ligase (NEB) unless otherwise stated. Transformation recovery was 60 min at 37 °C using 1 mL of pre-warmed SOC media. Transformation efficiency was $1.2e10^7$ CFU per mL recovery broth. The transformed cell library was used to inoculate duplicate cultures using 5 mL LB media supplemented with 50 ug/mL chloramphenicol. Shaking cultures were incubated overnight at 37 °C. One culture was used for cryostocking. Plasmid purification was performed on the other culture using an EZNA plasmid mini kit (Zymo). The plasmid library was sequentially digested with SpeI and EcoRI, followed by rSAP treatment. Digested plasmid pool was then purified via gel extraction.

To generate DNA templates for carrying out in vitro transcription and TSS mapping, the digested plasmid pool was ligated to a SpeI/EcoRI-digested DNA fragment containing a ribosome binding site (RBS) and mRFP1 coding sequence (Supplementary Fig. 10 and Supplementary Table 3), followed by purification and transformation into NEB 5-alpha Electrocompetent *E. coli* cells. The expression plasmid library (Supplementary Fig. 10) was then harvested from overnight shaking cultures grown in LB media supplemented with 50 ug/mL chloramphenicol. For quantification of library coverage, PCR was carried out on the plasmid library to generate promoter-barcode amplicons, followed by next-generation sequencing (Illumina MiSeq), barcode mapping, and promoter counting. 1 ug aliquots of the expression plasmid library were used as DNA template for the in vitro transcription reactions.

## in vitro transcription rate measurements

Replicate in vitro transcription reactions were carried out for 3 h at 37 °C, combining 1 ug of the expression plasmid library, 0.5 mM of each NTP, 1X *E. coli* RNA Polymerase Reaction Buffer, and 1 uL of *E. coli* RNA Polymerase Holoenzyme (NEB). DNA was removed by adding 2.5 units of TURBO DNase and incubating at 37 °C for 30 min. The RNA product was purified using a RNA Clean & Concentrator Kit (Zymo).

For quantification of RNA variant levels after transcription (RNA-Seq), cDNA first-strand synthesis reactions were first carried out on the harvested RNA, using 200 units of SuperScript IV (Invitrogen), 1X buffer, and transcript-specific primer 2 (Supplementary Table 3). PCR was carried out to generate barcode-containing amplicons (121 bp), using 5 ng of cDNA as template, primers 2 and 3, and 25 cycles of amplification (Supplementary Table 3), followed by next-generation sequencing (Illumina HiSeq 2500, 150 bp paired-end), barcode mapping, and barcode counting. For quantification of DNA variant levels in the expression plasmid library (DNA-Seq), PCR was first carried out on 10 ng of the expression plasmid library to generate promoter-barcode amplicons (847 bp long), using primers 1 and 2 and 25 cycles of amplification. The promoter-barcode regions were then condensed into shorter amplicons (271 bp) by treating the PCR product with T4 Polynucleotide Kinase (NEB), carrying out DNA circularization by ligation using T4 DNA Ligase (NEB), purifying the monomer DNA circles via gel extraction, and then PCR amplifying using primers 3 and 4

with 25 cycles (Supplementary Table 3). The promoter-barcode amplicon library was sequenced using Illumina HiSeq 2500 (150 bp, paired-end), followed by barcode mapping, promoter identification, and counting. Three independent in vitro transcription reactions, RNA-Seq, and DNA-Seq assays were carried out.

## Transcriptional start site mapping

Starting with RNA harvested from the in vitro transcription reactions, the locations of TSSs were determined by treating the RNA with TURBO DNase to remove template DNA, purifying, treating with RNA 5′ polyphosphatase (Epicentre) to remove 5′ phosphates, purifying, ligating the RNA to an 5′ RNA adapter (Supplementary Table 3) using T4 RNA Ligase (NEB), and purifying again. The treated RNA product was then converted to cDNA by treating with SuperScript IV (Invitrogen) and the transcript-specific primer 2 (Supplementary Table 3). PCR was then carried out using 10 ng of cDNA as template, primers 2 and 5 (Supplementary Table 3), and 30 cycles of amplification. The linear DNA product was then circularized by first treating it with T4 Polynucleotide Kinase, followed by ligating using T4 DNA Ligase. Monomer DNA circles were gel extracted. PCR was then carried out to generate the barcode-TSS amplicon library, using primers 3 and 6 in 25 cycles of amplification. The barcode-TSS amplicon library was then sequenced using Illumina HiSeq 2500 (150 bp, paired-end).

## Analysis of next-generation sequencing data

Barcode mapping, barcode counting, and promoter identification were carried out using custom software developed to analyze data from massively parallel reporter assays. The abundances of the cDNA-derived RNA transcript variants were quantified by counting the numbers of unique barcodes that were within 1 edit distance of the expected barcode sequences. The numbers of each DNA variant within the expression plasmid library were determined by first identifying associated barcode sequences that perfectly matched the expected barcode sequences, extracting the promoter sequence region, and mapping it to the expected promoter sequence according to barcode association. DNA counts were included if the promoter sequence perfectly mapped to the expected sequence. The RNA transcript counts and DNA counts were used to determine the relative transcription initiation rate of each promoter variant *i* in replicate *j* according to the formula:

$$r_{i,j} = \frac{\text{RNA}_{i,j}}{\text{DNA}_{i,j}} \times \frac{\sum_{i=0}^{n}\text{DNA}_{i,j}}{\sum_{i=0}^{n}\text{RNA}_{i,j}} \qquad (3)$$

where $\text{RNA}_{i,j}$ is the cDNA-derived barcode count for variant *i* in replicate *j*, $\text{DNA}_{i,j}$ is the DNA count containing promoter variant *i* in replicate *j*, and *n* is the total number of variants in replicate *j*. Replicate comparisons for RNA barcode counts, DNA promoter counts, and transcription rate measurements are shown in Supplementary Fig. 1. Transcriptional start site locations were determined using the TSS-barcode reads. For each read, the barcode was mapped and used to identify the associated reference promoter. The TSS location was then mapped by first identifying the position of a known constant flanking sequence, extracting the adjacent TSS-containing sequence, and carrying out a Smith-Waterman alignment using the associated promoter sequence as a reference. TSS locations were only counted when the TSS-containing sequence contained only one start location that perfectly matched the expected sequence.

## Data filtering

For model training, the set of promoter variants with a single, predominant transcription start site was defined by the intersection of three criteria: (i) promoter variants must have with at least 50 RNA-Seq and 50 DNA-Seq counts across all triplicate measurements; (ii) promoter variants must have one predominant TSS with at least twice the

counts as the next most predominant TSS; and (iii) promoter variants must have a predominant TSS within a 10 bp window surrounding the anticipated TSS location. Data filtering yielded 5193 promoter variants that satisfied these criteria. For model testing, the set of promoter variants with high-precision transcription rate measurements was defined using a single criterion: all promoter variants must have coefficients of variation (CVs) that are <0.40. CVs are calculated by determining the standard deviation of the transcription rate measurements and dividing by the mean of the transcription rate measurements across the three independent measurements. Data filtering yielded 5388 promoter variants that satisfied this criterion.

## A biophysical model of bacterial transcriptional initiation

Inside a cell or in vitro transcription reaction, free RNAP and σ-factor rapidly bind to form RNAP/σ complex. Whenever multiple σ-factors are present, they compete against each other to bind RNAP with the final concentrations of each RNAP/σ controlled by the σ-factor concentrations and their respective binding affinities. The RNAP/σ complexes then compete for binding to all available double-stranded DNAs in order to initiate transcription. This competition is enhanced as most copies of RNAP/σ are already bound to DNA and engaged in transcriptional elongation, leaving much fewer copies of RNAP/σ free for binding to DNA sites. Several factors enable us to assume that these binding interactions rapidly reach chemical equilibrium, including (i) the total amount of σ-factor is in excess as compared to the amount of freely available RNAP, leading to most free RNAP being bound by a σ-factor; (ii) the number of DNA binding sites is much higher than the number of freely available RNAP/σ complexes; and (iii) rapid RNAP turnover after transcriptional termination replenishes the supply of free RNAP and RNAP/σ complex. Accordingly, we can apply statistical thermodynamics to derive an equation that relates the transcription initiation rate of a DNA sequence region to the binding free energy of RNAP/σ. Equation 2 is the simplification of this equation, focused on predicting transcription initiation rates by RNAP/σ[70], which is the predominant RNAP/σ complex during exponential growth conditions. In our in vitro transcription assays, only one σ-factor (σ[70]) is present in the reaction, enabling us to precisely measure transcription initiation rates from RNAP/σ[70] binding events.

Consider a genetic system with $P$ copies of double-stranded DNA (either chromosomal or plasmid) and $R$ copies of freely available RNAP/σ[70]. On that DNA, the sequence region (upstream and downstream) around nucleotide position $i$ controls how well it binds to RNAP/σ[70] and initiates transcription. $P_i$ is the copy number of DNA at position $i$. $TX_i$ is the transcription initiation rate that begins producing a mRNA transcript at a TSS located at position $i$. Here, we formulate a model for predicting the transcription initiation rate in the forward direction ($TX^f_i$) given arbitrary DNA sequences surrounding position $i$. We then apply the same model on the reverse complement of the double-stranded DNA to predict the transcription initiation rate in the reverse direction ($TX^R_i$). As the model depends on non-symmetric amounts of upstream and downstream sequence centered on position $i$, the predictions $TX^f_i$ and $TX^R_i$ are two separate calculations.

We consider the process of prokaryotic transcriptional initiation at a single promoter as a two-state system. In state 1, RNAP/σ is not bound to the promoter DNA (free state). In state 2, RNAP/σ is bound to the promoter DNA with a stable transcriptional bubble, which includes the transition from the closed-to-open conformation, the initial melting of promoter DNA to create an unstable transcriptional bubble, and the formation of a stable transcriptional bubble (R-loop) via initial transcription of the ITR region. For many promoters, RNAP/σ does not engage in long-lived transcriptional pausing, enabling us to describe the system as having two-states. However, in a subset of promoters, it is possible to have long-lived intermediate states comprising of partially melted DNA bubbles or unstable scrunched R-loops, which could alter transcription rates. Here, the use of massively parallel

measurements does not provide the ability to measure the presence of intermediate states, preventing us from incorporating their presence in a model. Considering the thermodynamics of the two-state system, we compare the Gibbs free energy between the initial state (state 1) and the final state (state 2) to calculate the total change in the Gibbs free energy ($\Delta G_{total}$).

The chemical reaction and equilibrium condition for each DNA sequence region is therefore:

$$\begin{aligned} \text{RNAP} : \sigma^{70} + P_i &\Longleftrightarrow C_i \\ C_i &= RP_i \exp(-\beta \Delta G_{total,i}) \end{aligned} \tag{4}$$

where $R$ is the amount of available (free) RNAP/σ[70], $C_i$ is the number of promoter regions bound by RNAP/σ[70], $P_i$ is the copy number of DNA at position $i$, and $\Delta G_{total,i}$ is the total change in free energy of RNAP/σ[70] at a binding site that results in transcription with start site at position $i$. The total amount of available RNAP/σ[70] ($R_{total}$) is the sum of the free/unbound RNAP/σ[70] and the RNAP/σ[70] that is bound to DNA, which is:

$$R_{total} = R + \sum_j C_j = R[1 + \sum_j P_j \exp(-\beta \Delta G_{total,j})] \tag{5}$$

Which can be re-arranged to solve for $R$ in terms of $R_{total}$, giving:

$$R = \frac{R_{total}}{1 + \sum_j P_j \exp(-\beta \Delta G_{total,j})} \tag{6}$$

Substituting Eq. 6 into Eq. 4 and re-arranging, we obtain a relationship between $C_i$ and the total change in Gibbs free energy for RNAP/σ[70] at each potential DNA binding site in the genetic system.

$$C_i = R_{total} \frac{P_i \exp(-\beta \Delta G_{total,i})}{1 + \sum_j P_j \exp(-\beta \Delta G_{total,j})} = R_{total} \frac{P_i \exp(-\beta \Delta G_{total,i})}{Z} \tag{7}$$

Here, we see that the denominator in Eq. 7 is a partition function (labeled $Z$) that takes into account the competitive binding of RNAP/σ[70] to all potential DNA sites. When the genetic system is a bacterial cell, the summation to calculate the partition function $Z$ takes place over all chromosomal and plasmid DNA inside the cell. When the genetic system is an in vitro transcription assay containing thousands of plasmid DNA variants (each variant having a different promoter sequence), the summation takes place over all unique plasmid DNA variants. In both scenarios, the value of the partition function $Z$ is a large number that depends on the characteristics of the entire system (e.g., the cell strain, the growth conditions, the overall mRNA synthesis rate) and not the specifics of a single promoter sequence that is being altered.

The transcription initiation rate at the i[th] start site ($TX_i$) proportionally increases with the number of RNAP/σ[70] bound at that site ($C_i$), which is:

$$TX_i \propto R_{total} \frac{P_i \exp(-\beta \Delta G_{total,i})}{Z} \tag{8}$$

Measured TX rates for each dataset were normalized by promoter copy number using DNA-seq, removing the dependency of the model Eq. 8 on $P_i$. Rather than trying to predict, measure, or estimate $R_{total}$ and $Z$, we can instead carry out a comparative analysis by selecting one promoter from each dataset and labeling it as the reference promoter. The reference promoter is characterized under the same conditions as the studied promoters such that the number of free RNAP/σ ($R_{total}$) and partition function values ($Z$) are the same. The reference promoter's measured transcription initiation rate is labeled $TX_{ref}$ and has a constant value. The model's predicted total change in free energy for RNAP/σ[70] is $\Delta G_{total,ref}$. We then take the ratio between $TX_i$ and $TX_{ref}$ and

simplify, yielding:

$$\frac{TX_i}{TX_{ref}} = \exp(-\beta[\Delta G_{total,i} - \Delta G_{total,ref}]) \tag{9}$$

We then formulate a free energy model to calculate $\Delta G_{total}$ from sequence information. Our model considers each sequence-dependent interaction and then sums them together as linear, additive contributions. The free energy model is:

$$\Delta G_{total} = \Delta G_{-10} + \Delta G_{-10ext} + \Delta G_{-35} + \Delta G_{spacer} + \Delta G_{UP} + \Delta G_{disc} + \Delta G_{ITR} \tag{10}$$

where $\Delta G_{-10}$ is the contribution from the −10 hexamer sequence, $\Delta G_{-35}$ is the contribution from the −35 hexamer sequence, $\Delta G_{spacer}$ is the contribution from the DNA spacer's length and rigidity, $\Delta G_{disc}$ is the contribution from the discriminator sequence, $\Delta G_{ITR}$ is the contribution from the ITR sequence, and $\Delta G_{UP}$ is the contribution from the UP sequence.

The choice of a reference promoter is somewhat arbitrary, but it should have a well-measured transcription rate. Here, we selected a reference promoter with a measured transcription rate closest to the log-mean-center of the dataset. Without loss of generality, we also set $\Delta G_{total,ref}$ to zero. Therefore, the model is being trained to predict the differences in $\Delta G_{total}$ between the promoter variants and the reference promoter, which leads to an interaction energy scale that spans both negative and positive values. In general, interaction energies have negative values when they are stronger than the average interaction strength. Interaction energies have positive values when they are weaker than the average interaction strength.

### A non-linear free energy model to test for inter-motif cooperativity

We used the Utrecho et al. data for training the non-linear free energy model. For each promoter in the dataset, we used our linear free energy model to scan and predict the energy contribution for each of the promoter motifs considered in this study. These include (i) the UP element; (ii) the −35 hexamer; (iii) the spacer sequence, including the −10 extended; (iv) the −10 hexamer; (v) the discriminator; and (vi) the initially transcribed region. We then accounted for inter-motif interactions by extending Eq. 10 to include all pair-wise quadratic interaction terms between promoter motifs (interactions only, 6 × 5 terms). The model coefficients were determined using linear regression and the quadratic model was tested on all datasets (Tables S4–S6).

### Model training using machine learning

Machine learning was used to train, test, and validate several linear models (Ridge Regression, LASSO, and Elastic Net) and parameterize the unknown interaction energies. To do this, we first enumerated a list of sequence motifs and biophysical characteristics (Supplementary Table 1) that have the potential to contribute to the energetics of transcriptional initiation as quantified in our free energy model (Eq. 10). There are 472 features in the exhaustive unpruned list. For model training, we represented numerical features by using empirical data to calculate their values and normalizing each value by the dataset max (as to not exceed 1). Categorical features were represented using bit vectors (an array of 0 s and 1 s, the absence or presence of a feature). By doing this, both categorical and numerical features were brought to the same scale, and the relative energy contributions of each feature could be readily determined as the model coefficient multiplied by the feature value. We divided our filtered dataset (the set of promoter variants with a single predominant TSS, see the Data Filtering section above) into a training set (4673 promoters, 90%) and an unseen test set (520 promoters, 10%). Transcription initiation rate measurements were log-transformed and normalized by dividing each

promoter's transcription rate by the minimum transcription rate in the dataset.

We then carried out model training and hyperparameter optimization (alpha, 1 parameter) using tenfold cross-validation on the training set. The best models were then evaluated on the unseen test set. Accuracy metrics included the mean absolute error (MAE), mean squared error (MSE), and the squared Pearson correlation ($R^2$). Feature importance was then carried out using one-feature drop analysis. Model training and testing was repeated several times, each time dropping individual features or sets of related features. The differences in model accuracy were calculated and features were pruned if they did not appreciably increase model accuracy. Feature drop analysis identified several features whose values were highly correlated with other features in the dataset, for example, AT content in the UP region and the minor groove width in the UP region. The final set of pruned features was identified by including all features that yielded appreciable improvements in model accuracy. When faced with equivalent feature set choices, we selected the smaller set of features with fewer unknown coefficients, which often relied on biophysical calculations. We also compared each model's trained interaction energies against the canonical interactions known to have the most effect on transcription initiation rate, which we refer to as positive controls, and selected models that achieved 100% of these positive controls. Notably, the differences in interaction energies for the highest performing models were small, but the small differences could affect the rank-order of the top/bottom 10 sequence motifs, which affected their positive control successes. In Supplementary Table 2, we show the accuracies of the top performing linear models using either the full or pruned feature sets. We selected the Ridge Regression model for use in this study, though optimized versions of each linear model type yielded high accuracies on the unseen test set.

The final set of pruned features included: (i) The energetic contributions from binding to the −35 and −10 hexamer were represented using one-hot encoding of the presence of each possible 3 nt motif within each hexamer. All possible 3 nt motif sequences (64 3-mers) are included and their presence or absence is a categorical feature, totaling 256 features. (ii) The energetic contributions from the spacer region were decomposed into two sets of features, including the spacer length (15–20 bp, each represented as a categorical feature) and the spacer's DNA rigidity (a single numerical feature calculated based on the sequence-dependent persistence length of double-stranded DNA)[48]. (iii) The energetic contributions from binding to the −10 extended motif were represented using one-hot encoding of the presence of each possible 2 nt motif located upstream of the −10 hexamer, totaling 16 categorical features. (iv) The energetic contributions from the discriminator region were represented using one-hot encoding of each possible 3 nt motif in the first 3-nucleotides of the discriminator region, totaling 64 categorical features. (v) The energetic contributions from the UP region were represented using the numerical value of the calculated minor groove width in the distal and proximal UP sites (each 10 bp long)[45]. (vi) The energetic contributions from the ITR were represented by calculating the thermodynamic stability of the R-loop within the first 15 nucleotides of the ITR region (1 numerical feature), which is the free energy of the duplexed DNA-DNA complex subtracted by the free energy of the duplexed RNA-DNA complex. Numerical feature values are divided by the maximum possible value for normalization.

In addition, during the initial model training, feature encoding relied on the positional identification of sequence motifs using the following procedure. The predominant TSS was determined from TSS Mapping measurements. The promoter variant sequence was then scanned, taking into account that discriminator lengths varied from 5 to 10 nt and spacer lengths varied from 15 to 20 nt. The positions of the −10 and −35 hexamer motifs were then identified by scanning across the sequence, evaluating a position weight matrix tailored for each

motif, and calculating a motif score. The positions with the highest motif score were labeled as the −10 or −35 hexamer. If two or more locations had equivalent motif scores, the motif location with the most optimal spacer length (17 nt) was selected. This procedure was tailored to function alongside our experimental promoter library design, which included the 4096 possible −35 hexamer motif sequences alongside a consensus −10 hexamer as well as the 4096 possible −10 hexamer motif sequences alongside a consensus −35 hexamer. Even though these designs used a canonical spacer length of 17 nt, an alternate procedure that assumed that the spacer length was always a constant yielded scenarios where sequence changes led to shifts in the locations of the actual −10 or −35 motif and corresponding changes in motif sequence. Failing to account for these shifts led to mismatched inter-action energies during model training, for example, non-canonical motifs having interaction energies similar to canonical motifs, created by the formation of a canonical motif with a shifted location.

### Individual promoter construction and Isogenic characterization

Individual promoters were constructed to test the effects of changing UP and ITR regions on transcription rates (Fig. 4) and to test the automated design of promoters using model predictions (Fig. 5). Promoters were constructed by synthesizing pairs of overlapping oli-gonucleotides and using PCR assembly to create insertion DNA cas-settes. Cassettes and the mRFP1-expression plasmid pFTV1 were double-digested, purified via gel extraction, ligated, and transformed. Plasmid products were sequence verified using Sanger sequencing. Isogenic *E. coli* DH10B cells were transformed with each plasmid, fol-lowing by quantification of their mRFP1 expression levels during exponential growth, using spectrophotometry and flow cytometry measurements. Cells were grown overnight in LB media supplemented with 50 ug/ml chloramphenicol, followed by a 1:100 dilution into 200 ul M9 minimal media supplemented with 50 ug/ml chlor-amphenicol within 96-well optical bottom microtiter plates. OD600 and mRFP1 fluorescence measurements were recorded every 10 min until cultures reached an OD600 of 0.20. Cultures were then serially diluted 1:10 into pre-warmed supplemented M9 media and grown again until they reached an OD600 of 0.20. 10 ul aliquots were then extracted and added to 190 ul PBS with 2 mg/mL kanamycin to stop protein production. Flow cytometry was then carried out on the fixed cells to record their fluorescence distribution (BD LSR Fortessa). The mRFP1 fluorescence level was the arithmetic mean of the measured fluorescence distribution subtracted by autofluorescence, which was the arithmetic mean of the fluorescence distribution of wild-type *E. coli* DH10B cells. An example of our gating strategy is provided in Sup-plementary Fig. 14. All designed promoter sequences, model calcula-tions, and flow cytometry measurements are available in the Supplementary Data 1.

### Monte Carlo analysis of promoter genetic context

The core promoter sequences for the Anderson Library were retrieved from the iGEM website. We randomly generated 10,000 pairs of 30 bp DNA sequences with equal weighting of all four nucleotides. These DNA pairs were used to flank the core promoters creating a simulated DNA context. The Promoter Calculator was then used to scan and predict the transcription rate of each member of the promoter library in all 10,000 contexts. Box plots were generated using each promoter's predicted transcription rate distribution.

### Automated promoter sequence design

We used an optimization algorithm (simulated annealing) to auto-matically design 35 promoter sequences with systematically varied transcription initiation rates, using the statistical thermodynamic model prediction within the optimization algorithm's objective func-tion. Designed promoter sequences were allowed to have variable TSS positions and use the surrounding upstream/downstream DNA

sequence as part of the promoter. The surrounding upstream/down-stream DNA sequences were kept constant for all promoter designs. The promoter lengths varied from 60 to 77 bp, yielding an average sequence similarity of only ~27%.

### Identification of cryptic promoters for genetic circuit debugging

The statistical thermodynamic model was used to predict the tran-scription initiation rates across the 11-promoter genetic circuit (6793 bp), as studied and characterized in Borujeni et al.[8], and identify desired and undesired/cryptic TSSs on both its sense and anti-sense DNA strands. We labeled a location as a TSS when its predicted tran-scription initiation rate was at least threefold higher than the average transcription initiation rate per each strand. We used RNA-Seq mea-surements and RNAP flux inferences from Borujeni et al.[8] to assess the accuracy of these TSS predictions (Fig. 6). To do this, we counted the number of actual TSS peaks that were within 10 bp of the model-predicted TSSs and divided by the total number of model-predicted TSSs. This window threshold is reasonable considering that RNA-Seq measurements using 150 bp reads have limited positional precision and the RNAP flux inference calculation was shifted by up to 20 bp for known promoters with predominant TSSs.

### Model benchmarking

We carried out benchmarking by comparing transcription rate mea-surements to model predictions, using our model and the models developed by Lagator et al.[15] Predictions on promoters using the Lagator models were carried out by using their provided source code and data files, which were retrieved from https://github.com/szarma/Thermoters. We show all Lagator model predictions on their training and test datasets (Supplementary Fig. 11) as well as the training and test datasets used in this study (Supplementary Fig. 12). In the Lagator et al. manuscript, comparisons are made between $\log_{10}(P_{on})$ and $\log_{10}(TX)$, where $P_{on}$ is the Lagator model prediction and TX is the measured transcription rate. In our model formulation, comparisons are made between the predicted $\Delta G_{total}$ and $\log(TX)$. To compare between models, we converted Lagator predictions to natural log scale. Both models assume a linear relationship between these predicted and measured quantities. We therefore calculated the Pearson $R^2$, the MAE, and the MSE for each model on each dataset (Supplementary Table 4, 5 and 6). Unlike the Pearson $R^2$, the MAE and MSE are affected by the slope and intercept of the linear relationship between the predicted and measured quantity. We therefore carried out three approaches to calculate the MAE and MSE for all models on all datasets: (1) for each model and dataset, we identify the best-fit slope via linear regression and use it to calculate MAE and MSE (Supplementary Table 4); (2) for each model and dataset, we identify the best-fit intercept via linear regression and use it to calculate MAE and MSE (Supplementary Table 5); and (3) for each model and dataset, we identify the best-fit slope and intercept via linear regression and use it to calculate MAE and MSE (Supplementary Table 6). We also compared model residuals for the best performing models with only a proportionality constant (slope-only, Supplementary Fig. 13) to test the proportionality assumption both models operate under. *F*-tests were carried out to compare model errors (Supplementary Data 4).

### Reporting summary

Further information on research design is available in the Nature Research Reporting Summary linked to this article.

## Data availability

All promoter sequences, model calculations, and experimental mea-surements are available in Supplementary Data 1, 2, 3, 4 and 5. Next-generation sequencing read data files in fastQ format are publicly available at NCBI with accession identifier PRJNA754118.

## Code availability

We used Python v3.7.7 and custom code that employed the following Python modules to develop Promoter Calculator: sklearn v1.0.2, SciPy v1.7.3, numpy v1.21.2, pandas v1.3.5, matplotlib v3.5.0, pickle v4.0. Flow cytometry analysis was done with FlowCal 1.2.2. A Python source code implementation of the statistical thermodynamic model of transcriptional initiation is available at https://github.com/hsalis/SalisLabCode with Git tag name r2022.

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

## Acknowledgements

This project was supported by funds from the Defense Advanced Research Projects Agency (HR001117C0095), the Department of Energy (DE-SC0019090), and the National Science Foundation (MCB-2131923). T.L.L. was supported in part by the National Institutes of Health CBIOS training program (1T32GM102057).

## Author contributions

T.L.L., A.H., and H.M.S. conceived the study. T.L.L. designed and carried out the experiments. T.L.L., A.H., and H.M.S. developed the algorithms and performed the data analysis. T.L.L. and H.M.S. wrote the paper. All authors read, edited, and approved the paper.

## Competing interests

H.M.S. is a founder of De Novo DNA. T.L.L. and A.H. declare no competing interests.
