## [Peer Review File · Nature Communications]

Reviewers' Comments:

Reviewer #1:

Remarks to the Author:

This manuscript presents a statistical thermodynamic model that predicts gene expression levels from constitutive promoter sequences based on the binding of sigma70-RNA polymerase complex, and then shows how such a model can be utilized to guide the design of synthetic promoters. The paper has some interesting novel insights, in particular: it uses machine learning to develop a model that links to the known and understood mechanisms of promoter function, rather than utilizing a 'black-box' approach; it characterizes the independent contribution of each component of a promoter to overall expression in a systematic manner; and it demonstrates the utility of such models for optimization of synthetic promoters. However, I have two major concerns about the manuscript that relate to its generalizability to novel data sets and the overall novelty of this work.

The major question of novelty and generalizability of this work stems from the fact that several models based on statistical thermodynamics, which are capable of generating accurate predictions of gene expression levels from sequences, already exist.

While the approaches are different in their formulation of the underlying mechanisms of regulation, fundamentally they all focus on the energy contribution of specific positions within the promoter on the overall binding energy of RNA polymerase to DNA (See Kinney et al. 2010 and Lagator et al. 2022 for some examples). Furthermore, just like the other models of this type, this model also focuses on the sigma70 constitutive promoters, as opposed to any other sigma factors or regulated promoters. This manuscript does an excellent job of creating a large number of promoter variants and has expanded our understanding of how mutations impact specific promoter components (like the UP elements and ITR) beyond previous works. However, it is limited by those variants being constrained to a specific section of the promoter. For example, all -10 hexamer variants are introduced into a fixed genetic context, with a consensus -35 hexamer and a fixed, 17bp sequence. The problem with this approach is that, while exhaustive in a local sense, it limits the model to a specific mutational neighbourhood and hence limits its generalizability. Einav & Phillips 2019 PNAS showed that the interactions between -10 and -35 hexamers can have a large effect on expression levels. The design presented in this manuscript cannot account for these effects, meaning that at least one known major contributor to gene expression levels is not accounted for. Especially in the context of other existing models (Einav & Phillips, Lagator et al.), it is not clear to me that the focus on individual components of an already pretty well described system, as opposed to the interactions between different components of the promoter which are poorly understood, is standing out as novel.

Furthermore, the focus on what goes on within each component while the context is fixed means that it is not clear from the presented manuscript that this model significantly outperforms the existing models. For example, the authors present a comparison to a recent paper by Lagator et al., and discuss that their model performs better on two independent datasets. However, their model performs very poorly on the datasets generated by that paper, and especially on the fully random mutant library (36N) where their model has an R2 of 0.15. This poses a serious question about the generalizability of the model, which is not helped by the fact that the authors do not discuss the libraries in which their model performs poorly but rather focus only on those examples where it performs better.

I suspect that the two issues – of novelty and of generalizability – are linked, because the outstanding and important advances in the prediction of promoter function I believe stem from understanding how different components of a model interact with each other. Together, these concerns potentially make the audience of this paper more specialized, as the novelty is contained to synthetic biology applications (Fig.5 and 6) and the evaluation of less important contributors to expression levels (like the ITR and the UP element) (Fig.1D and 2C).

Detailed comments:

It would be very useful to provide a breakdown of MSE of the presented model based on the general expression level being predicted. What I mean is, what is the MSE for promoters that have

high expression level, vs those with intermediate, vs those with low. This would help understand if the model performs equally well across the entire expression range, or if it underperforms at some expression level ranges. If the latter is the case, it would be important to explore why that might be, especially in the light of various mechanisms of promoter function not included in this model, such as those explored by Einav and Phillips or Lagator et al.

It would be really important to understand why the presented model performs poorly on the fully random promoter library from Lagator et al. (the 36N library). The low performance on random sequences strongly suggests that complex interactions based on genetic context of the promoter play a large role, at least while a promoter is defined as leading to a single TSS. Fig.5 addresses some of these questions, but in a very limited way using only 16 mutants. Providing deeper insights into what happens when a promoter moves from a given mutational neighbourhood to which their model is optimized and into random sequence space would be critical to increase the novelty of this work.

Another good test in Fig.5 would be to design more promoters predicted to have same expression levels but very divergent sequences, to evaluate how repeatable the predictions are.

Legends to some of the figures and tables could be expanded to give more details. For example, Tables S4,5,6 – what do the colours indicate?

I think one of the biggest contributions of this paper is how they used ML to derive model parameters that were comprehensible mechanistically. Expanding a bit in the main text on how this was done would be useful.

In the -10 binding motif, the authors examine the role of mutations by looking at every possible 3-mer. Energy Matrix-based models look at each individual position. I am not sure what the difference is between these two ways of modelling binding. For example, does the 'individual' position approach linearly add up to the observed effects of 3-mers? If not, when does it step away and why? Doing such an analysis can result in key insights into the interactions between mutations within each promoter component.

What is the distribution of spacer lengths in the experimental dataset, are they evenly distributed across all 5 relevant lengths?

How long are the UP and ITR sequences used in the 'Promoter Context Effects' section? It's possible that their relatively low effect on model accuracy is due to their short length, meaning that what 'context' is might be longer than the varied UP and ITR elements. Therefore, it would be good to note their exact length.

The introduction mentions the word 'interactions' quite a bit, but as discussed above, it is unclear which interactions are actually studied. In particular, given that the experimental design has a very limited exploration of the interactions between promoter components (what I think is the most important area where we lack knowledge), I would avoid using this word.

Authors claim 'a key novelty of our model is its complete set of interaction energies, covering all possible promoter sequences...' – It would be important to clarify what the word 'interaction' means here, because it applies to only a small number of potential interactions between promoter components.

The 'model generality' section would be easier to follow if the authors referred to the external datasets by the author names, as opposed to just the reference, as it is difficult to go back and forth to the reference section to recall that 16 is Urtecho et al. and 12 is Hossain et al. etc.

Reviewer #2:
Remarks to the Author:
Summary:

In the manuscript by LaFleur et al. ("Automated Model-Predictive Design of Synthetic Promoters to Control Transcriptional Profiles in Bacteria"), the authors create a biophysical model of transcription initiation of sigma -70 promoters in *E. coli* and train their model using high throughput assays and machine learning. Starting with 472 biophysical parameters, they train and test their model on 5193 synthetic promoter variants and use the resulting data to prune their model to 346 parameters. With this model, they characterize the impacts of individual promoter motifs on transcription initiation. The authors test their model against a previously published model using their own in vitro dataset, as well as the Urtecho and Hossain datasets. The authors show that the new model yields significant improvements in predicting transcription rates for synthetic promoters ($R^2 = .79, .60, \text{ and } .45$ respectively for the new model; vs. $R^2 = .60, .45, \text{ and } .39$ for previously published model). They also demonstrate the applicability of their model for gene circuit debugging and the generation of de novo promoters with desired transcription rates. This model, accessible as an online software tool, is well-suited for synthetic biologists looking to reliably predict and control transcription rates of their genetic constructs. This work advances the field of genetic circuit design by combining a biophysical model for transcription initiation with machine learning informed by massively parallel in vitro assays. This is a growing area of research and this is nice work that advances the state-of-the-art. The writing and figures are well composed and easy to understand. One of the following comments are addressed, it should be suitable for publication in Nature Communications.

Major Comments:

1. The authors seem to sweep under the rug the fact that of their initial 14206 promoters, only 5193 are used due to the recurring presence of a cryptic TSS. It would be helpful for the authors to explain the impact of the cryptic promoter within the ORF more explicitly. According to Fig 1C, this cryptic TSS looks to be consistently around position 105. Would the inclusion of promoters with cryptic start sites in the model training help inform parameters and motifs that more likely lead to a cryptic TSS? Could this improve biological relevance for predicting transcription rates of endogenous promoters and promoter design?
2. Additionally, what is the composition of the 5193 promoters used? Is the set skewed to variations within one of the promoter regions? It seems like this could skew the retained features and decrease the generalizability of the model. For instance, Table S1 shows all -10 and -35 3-mers were retained, but no ITR features were retained. What is the explanation for this, and how are the Gibbs free energy contributions calculated when no features are retained?
3. Regarding the model, the connection between the 346 features to the different Gibbs free energy contributions is not clear, especially the categorical properties such as spacer length. This information seems crucial for a biophysical model and should be included to strengthen the manuscript. Additionally, it is not obvious how to interpret and compare the different interaction energies in Figure 2E, have they been normalized? For instance, is a strong -35 interaction equal to a strong -10 interaction?
4. The authors tried a linear combination of different features to estimate interaction energies, and a linear combination of these interactions energies to estimate transcription rates. The manuscript would be enhanced by rationale for why this approach was chosen and a more in-depth discussion of the limitations when compared to other approaches such as non-linear regression or neural networks.
5. The authors do a good job by benchmarking different models and datasets, but it is hard to interpret the comparisons without stronger statistical tests, especially given the different number of parameters across models. The use of F-test or information criterion (AIC or BIC) to compare the models would help strengthen the author's claims.
6. Page 9, "However, most promoters do not contain canonical motif sequences; their TX rates are controlled by a mixture of weaker interactions. A key novelty of our model is its complete set of interaction energies, covering all possible promoter sequences, which provides the ability to predict the TX rate of any $\sigma 70$ promoter." The claim that their model covers "all possible promoter sequences" is misleading, especially given that only $\frac{1}{3}$ of designed promoters were used in the training of the model. Additionally, other models (Urtecho et al.) can also predict transcription rate

of any $\sigma 70$ promoter.

7. While the exclusion of alternative sigma-factors in the in vitro assay helps illuminate the “true” transcription rate due to $\sigma 70$ -RNAP interactions, it does come at the expense of biological relevance and potentially limits the generalizability of this model for the prediction of transcription rates and promoter design. This is shown by the model’s low accuracy on in vivo datasets. This limitation should be discussed more explicitly.

Minor Comments:

1. Discussion about challenges to generalizing this work to other organisms and other sigma-factors would be valuable.

2. In Fig 1D, the range transcription rates of each motif library is shown. The “all” bar may be misleading, as this represents the greatest difference between a single variant in one motif and a single variant in another motif. A total range of 133 does not cover the wide range of transcription rates in the genome. The model may behave poorly at the extremes (very low transcription or very high transcription), and there is no data or much discussion to address this point.

3. “We next evaluated whether these genetic context effects are potentially widespread by carrying out Monte Carlo simulations to predict the range of TX rates expected when varying the UP and ITR regions of several commonly used promoters (Figure 4B). Promoters with lower average TX rates exhibited higher sensitivity to changing the UP and ITR regions with an overall average coefficient of variation of 0.40.” Cannot tell this statement is true based on Figure 4B, there is no statistical significance test. Moreover, there is no methods section for Monte Carlo simulations.

4. Page 5, “Here, we arbitrarily set $DG_{total,ref}$ to zero so that stronger (weaker) interaction free energies have more negative (positive) values in comparison.” This sentence is hard to follow.

5. Readability of Fig 6 is difficult. Axes for 6a can be standardized. Putative start sites should be labeled. Additionally, it is unclear where the cut off for a TSS comes from in 6a.

6. How does a 55% prediction accuracy for identifying cryptic promoters compare to other methods?

7. Why is the train-test split 90-10 when figure 2A shows minimum error at $\sim 25\%$ training set size?

Title: Automated Model-Predictive Design of Synthetic Promoters to Control Transcriptional Profiles in Bacteria

Authors: Travis L. LaFleur¹, Ayaan Hossain², Howard M. Salis^{1,2,3,4} *

NCOMMS-22-08328

We would like to express our deep appreciation for the reviewers' constructive comments, which pointed out several areas where the manuscript could be improved. Below, in our point-by-point response, we answer the reviewers' questions and describe **additional experimental datasets** and **manuscript modifications** to further support our conclusions. The reviewers' questions are listed below, followed by our responses (**blue text**) and quoted manuscript revisions (**green text**).

Reviewer #1:

This manuscript presents a statistical thermodynamic model that predicts gene expression levels from constitutive promoter sequences based on the binding of sigma70-RNA polymerase complex, and then shows how such a model can be utilized to guide the design of synthetic promoters. The paper has some interesting novel insights, in particular: it uses machine learning to develop a model that links to the known and understood mechanisms of promoter function, rather than utilizing a 'black-box' approach; it characterizes the independent contribution of each component of a promoter to overall expression in a systematic manner; and it demonstrates the utility of such models for optimization of synthetic promoters. However, I have two major concerns about the manuscript that relate to its generalizability to novel data sets and the overall novelty of this work.

Our work contains highly novel results that push forward the state-of-the-art in three distinct areas. We list them below before we dive deeper into the reviewers' comments.

Key novelty #1: Our work is the first to carry out massively parallel and fully *in vitro* transcription assays and transcriptional start site mapping on thousands of designed promoters with the overall objective of **precisely measuring** RNAP/ σ^{70} -DNA interaction strengths. A good model depends on collecting a good dataset; our dataset has several characteristics that represent a **transformative improvement** in the state-of-the-art. **The first key advantage of our dataset is that the *in vitro* transcription assays eliminate the presence of several confounding variables.** For example, because there are no RNases in our *in vitro* transcription assays, changing the ITR sequence (part of the mRNA) will have no effect on mRNA stability or mRNA decay in our dataset. Prior work has shown that the first 3 nucleotides of the mRNA transcript can affect mRNA levels by over 10-fold by changing the transcript's decay rate [Cetnar & Salis. *ACS Syn Bio.* (2021)]. There are also no ribosomes in our *in vitro* assay so we do not worry about unintended changes in translation rate or coupling between transcription & translation rates (unlike *in vivo* measurements). Because we only add RNAP/ σ^{70} to our *in vitro* assay, we also know with certainty that the alternative σ -factors do not contribute to the promoters' transcription rates. And because there are no cells, we have eliminated all growth-coupled bias in our measurements, for example, when very strong promoters produce so much RNA or protein that they inhibit cell growth, which greatly affects DNA-Seq and RNA-Seq read counts. Finally, because we carried out transcription start site

mapping, we can conclusively identify the actual motifs that form contacts with RNAP/ σ^{70} . Without transcriptional start site mapping, promoters with multiple RNAP/ σ^{70} binding sites could skew the training of the biophysical model.

The **second key advantage of our dataset** is that we designed the promoter sequences to systematically perturb and vary the known interactions that affect σ^{70} -dependent transcription rate; notably, our dataset is the first to measure RNAP/ σ^{70} interactions using all possible -10, -10 extended, and -35 motif sequences and it is the first to systematically vary the biophysical characteristics of the UPS and ITR elements. Our promoter designs encompass several strategies. Yes, we characterized several sequence sets that use a “change only one motif at a time” strategy to precisely measure their effect on transcription rate. But we also characterized many sequence designs where we varied 2 or more motifs in combination to determine the synergistic effect (e.g. we combined several designed UP elements with canonical vs. anti-canonical hexamers). These designs focused on known synergism between different binding regions (UP, hexamers, -10 extended). This distinction is in sharp contrast to prior datasets where promoter sequences were randomly generated (either fully random or randomized starting from a strong promoter as in Lagator et al.) or where sets of only 8 motifs were combined to generate promoters (as in Urtecho et al.).

The **third key advantage of our dataset is that our experimental workflow achieved high precision** with a high standard of excellence, due to our usage of designed (not UMI randomized) barcode sequences to distinguish promoter designs, our next-generation sequencing with very high read depth as compared to prior efforts, and carrying out the experiments in triplicate to quantify and assess our precision. Many prior datasets (particularly using Flow-Seq assays) do not carry out any replicates and therefore can not be assessed for reproducibility and precision.

To clarify, when other datasets use UMI randomized barcodes, it is incorrectly assumed that distinct barcode sequences correspond to distinct promoter designs; but, due to NGS base calling error and PCR bias, the observed reality is that reads mapped to a UMI barcode often point to multiple different promoter designs, while several distinct promoter designs may become associated with the same mapped UMI barcode. These “barcode collisions” can greatly distort the observed transcription rates. For example, a strong promoter’s UMI barcode can “bleed into” a weak promoter’s UMI barcode whenever those UMIs have similar (not the same) sequences. Further, with completely randomized barcodes (20N), a minimum read depth greater than 4^{20} is required to see every barcode once. This phenomenon disproportionately effects weak promoters, causing a lack of resolution at the low-TX range. These types of confounding interactions are not discussed much in the literature. **We avoid these barcode biases altogether** by designing defined barcode sequences for each promoter sequence (1 to 1) while maximizing pairwise Hamming distance between barcodes so they remain distinguishable even when subjected to NGS base calling error and PCR bias error.

Key novelty #2: Our work is **the first** to apply a biophysical model of transcriptional initiation to **forward-design completely non-natural & dissimilar full-length promoter sequences with desired transcription rates**. This capability is highly demanded by Synthetic Biologists who would like to control and optimize their transcription rates for biotech applications. To stringently test our design algorithm, we ensured that our designed promoters had highly dissimilar sequences

so that all motif sequences were being varied at the same time. This type of test is a good way of investigating whether synergistic interactions between motifs are important to controlling transcription rates. *In response to the reviewers' comments, we designed and characterized even more non-natural promoters* to further strengthen this conclusion (**Figure 5** and described in detail below). We also designed and tested **DNA sequences with minimized transcription initiation rates** to demonstrate how the model can be used to remove internal promoters within other genetic parts (**Figure 6C**). Further, our algorithm is **the first** to ensure that **designed promoter sequences only contained a single, predominant transcriptional start site** to provide even more tunable control over transcription rate. To be clear, the model developed in Lagator et al. was never used to design a promoter sequence, which is a significant shortcoming. The model developed in Urtecho et al. does not have the ability to design synthetic promoters outside the limited set of motifs used in their dataset (their model can not predict the transcription rate of arbitrary input DNA sequences). Prior efforts have also developed classification models (not regression models) to label a promoter's strength across 12 possible outcomes [Van Brempt et al. Nat Comm 2020]. This model was used to vary the spacer sequence composition with a fixed spacer length within a background sequence with constant hexamer motifs, resulting in promoter sequences that varied transcription rate by about 20-fold. Explicitly, their sequence constraint is:

```
[GGTCTATGAGTGGTTGCTGGATAAC] [TTTACG] [NNNNNNNNNNNNNNNN] [TATAAT] [ATATTC] [AGGGAGAGCACAAAGGTTTCCTCTACAAATAATTTTGTAACTTT]
```

where Ns denote the spacer positions that were designed by their algorithm. To compare, our model & design algorithm can design the entire promoter sequence with varied spacer lengths, which resulted in dissimilar promoter sequences that varied transcription rate by 1525-fold.

Key novelty #3: As we show in our benchmarking, **our biophysical model of transcriptional initiation is more accurate than all prior sequence-complete models** and takes into account **more RNAP/ σ^{70} interactions** than all prior sequence-complete models. The definition of a sequence-complete model is one that can carry out a prediction on arbitrary inputted DNA sequences. The model described in Lagator et al. is sequence-complete, but the ones described in Urtecho et al. and Van Brempt et al. are not. Our biophysical model takes into account the effects of the UP, discriminator and ITR elements whereas the model developed in Lagator et al. does not. In our work, we showed that changing only the UP and ITR elements can affect transcription rates by a significant amount (**Figure 1**, up to 21-fold), which makes their inclusion into a model a high priority, particularly when using the model to design non-natural promoter sequences. Lagator et al. is a recent publication and our work was carried out independently. We use very distinct approaches in developing and testing our models.

We would like to note that, when carrying an examination of the Lagator et al. model source code and using it to run our benchmarking study (in the original manuscript), there were inconsistencies between what is described in their article vs. the calculations that are actually performed in their source code. For example, for each dataset, the reported R^2 values in Lagator et al. were determined through an iterative process whereby multiple external parameters were varied until the highest R^2 value is achieved. These external parameters include a “detection threshold” (a filter that fixes the minimum predicted transcription rate to a constant value, defined by what maximizes R^2), a “chemical potential” (an arbitrary number that shifts the prediction scale up or down regardless of the growth media), and an arbitrary slope and intercept that relates their predicted $\log_{10}(P_{ON})$ to the measured $\log_{10}(\text{transcription rates})$. In principle, it could be valid to utilize a detection threshold based on experimental parameters, but their actual detection threshold values are varied greatly

between datasets (well above a realistic threshold) to give the effect of excluding many outlier predictions. In principle, a “chemical potential” number reflecting changes in RNAP/ σ concentration could be valid, but in practice, they arbitrarily chose (to maximize R^2) different chemical potential numbers even though some datasets used the same strains and were grown in identical conditions. Because “seeing is believing”, we show the predictions of the Lagator et al. model without the use of a “detection threshold” and setting the “chemical potential” to zero while utilizing their Extended Model. These plots are also included in our Supplementary Information. The Pearson R^2 values for each plot are shown in the legend. Significant data clustering can inadvertently increase Pearson R^2 values regardless of the linearity of the comparisons. A better accuracy metric is the mean absolute error (MAE) or the mean squared error (MSE), which we include in our benchmarking.

Figure S11: Lagator Extended Model Predictions on Lagator Datasets (A) Lagator extended model predictions for 2903 PI mutant promoters are compared to reported *in vivo* Sort-Seq means. R^2 is equal to 0.73. (B) Lagator extended model predictions for 12194 Pr mutant promoters are compared to reported *in vivo* Sort-Seq means. R^2 is equal to 0.67. (C) Lagator extended model predictions for 11523 random 36N sequences are compared to reported *in vivo* Sort-Seq means. R^2 is equal to 0.44. All data is available in **Supplementary Data 3**.

Figure S12: Lagator Extended Model Validation (A) Lagator extended model predictions for 5391 designed promoters (this study) are compared to *in vitro* transcription rate measurements. R^2 is equal to 0.60. (B) Lagator extended model predictions for 10898 genome-integrated promoters are compared to *in vivo* transcription rate measurements. R^2 is equal to 0.45 (C) Lagator extended model predictions on 4350

non-repetitive plasmid-encoded promoters are compared to *in vivo* transcription rate measurements. R^2 is equal to 0.39. All data is available in **Supplementary Data 3**

In the original submission, we carried out a comprehensive model comparison across several datasets (**Supplementary Table S4**), calculating the Pearson R^2 , the mean absolute error (MAE), and the mean squared error (MSE) for all comparisons. In all calculations, we do not apply a variable “detection threshold” and we do not add an arbitrary number (a “chemical potential”) to model predictions. We rely only on a key assumption made by both us and Lagator et al.; that predicted values are directly proportional to log-transformed expression levels. Therefore, for all models (including from Lagator et al.), we use the best-fit slope to calculate the MAE and MSE. This is the best approach for an apples-to-apples model comparison. We copy **Supplementary Table S4** below for your convenience.

The benchmarking revealed that our linear model (trained on our novel *in vitro* dataset using machine learning) was **more accurate (higher R^2 , lower MAE, lower MSE) when applied to the Urtecho, Hossain, and LaFleur datasets** than the Lagator model (in either its standard or extended versions). Even when our linear model is applied to the datasets collected in Lagator et al. (the Pl, Pr, 36N promoter datasets), **our model achieves a lower MAE and a lower MSE** than the models developed in Lagator et al. This is important as, even though their model achieves a higher R^2 on their own datasets, it does so from high clustering at the extremes of the range.

Table S4: Model Benchmarking – Slope Only

	LaFleur (linear)	LaFleur (quadratic)	Lagator (standard)	Lagator (extended)	N samples
R² Urtecho^a	0.60	0.69	0.47	0.45	10,898
MAE Urtecho^a	0.93	0.56	1.32	1.33	
MSE Urtecho^a	1.28	0.59	3.93	3.99	
R² Hossain^a	0.45	0.46	0.42	0.39	4,350
MAE Hossain^a	1.08	1.96	4.83	4.85	
MSE Hossain^a	1.88	5.66	31.5	31.0	
R² LaFleur	0.79	0.76	0.58	0.60	5,391
MAE LaFleur	0.32	0.79	1.43	1.56	
MSE LaFleur	0.17	0.88	3.98	4.27	
R² Lagator Pl	0.47	0.49	0.67	0.73	2,903
MAE Lagator Pl	0.91	0.98	3.53	3.74	
MSE Lagator Pl	1.34	1.70	14.5	16.1	
R² Lagator Pr	0.35	0.37	0.55	0.67	12,194
MAE Lagator Pr	0.51	1.06	3.14	3.40	
MSE Lagator Pr	0.62	1.79	13.1	14.7	
R² Lagator 36N	0.19	0.15	0.33	0.44	11,523
MAE Lagator 36N	0.59	1.71	1.16	1.34	
MSE Lagator 36N	0.57	4.50	2.31	3.00	

^aUnseen datasets to both models during training. See Discussion for details.

In the updated manuscript, we extended the model comparison in the Supplementary Information by providing residual distributions for the benchmarking of our linear model and the extended model developed by Lagator et al. (**Figure S13**). Despite the Lagator model being trained on the Pr (11% of training data), PI (46% of training data) and 36N (43% of training data) datasets, we see that their model's **raw predictions** are **highly biased** to the 36N dataset. Under the proposed assumption of proportionality made by both models, this makes the utility of our model greater, as **raw model outputs are closer to the reality of experimentally observed expression levels**.

Figure S13: Slope-only Model Benchmarking, Residual Distributions (A) Model residual distributions for Urtecho et al. (left), Hossain et al. (middle) and LaFleur et al. (right). (B) Model residual distributions for the PI dataset from Lagator et al. (left), Pr dataset from Lagator et al. (middle) and 36N dataset from Lagator et al. (right). Blue distributions are from the LaFleur et al. linear model, red distributions are from the Lagator et al. extended model. Model residuals are the absolute difference between the model prediction [ΔG_{total} for LaFleur and $\log(P_{\text{on}})$ for Lagator] and the natural log of the measured transcription rates, using a best-fit slope relationship for each model and dataset.

While the approaches are different in their formulation of the underlying mechanisms of regulation, fundamentally they all focus on the energy contribution of specific positions within the promoter on the overall binding energy of RNA polymerase to DNA (See Kinney et al. 2010 and Lagator et al. 2022 for some examples). Furthermore, just like the other models of this type, this model also focuses on the sigma70 constitutive promoters, as opposed to any other sigma factors or regulated promoters. This manuscript does an excellent job of creating a large number of promoter variants and has expanded our understanding of how mutations impact specific promoter components (like the UP elements and ITR) beyond previous works. However, it is limited by those variants being constrained to a specific section of the promoter. For example, all -10 hexamer variants are introduced into a fixed genetic context, with a consensus -35 hexamer and a fixed, 17bp sequence. The problem with this approach is that, while exhaustive in a local sense, it limits the model to a specific mutational neighbourhood and hence limits its generalizability. Einav & Phillips 2019 PNAS showed that the interactions between -10 and -35 hexamers can have a large effect on expression levels. The design presented in this manuscript cannot account for these effects, meaning that at least one known major contributor to gene expression levels is not accounted for. Especially in the context of other existing models (Einav & Phillips, Lagator et al.), it is not clear to me that the focus on individual components of an already pretty well described system, as opposed to the interactions between different components of the promoter which are poorly understood, is standing out as novel.

We agree that many of our promoter designs use a “change one motif at a time” strategy with the explicit goal of precisely measuring the interactions at that motif. However, we do have several (over 1000 in the training data) of designs where we combine multiple motifs together in different combinations (e.g. varying the biophysical parameters of the UP element and using either canonical or anti-canonical -10 and -35 hexamers) to observe the effects. We agree with the reviewer that our *in vitro* dataset (on its own) does not have sufficient breadth to fully test all possible combinations of interactions. This is the key reason why we (in our first submission) carried out a model benchmarking analysis using many datasets with distinct sequence compositions and experimental testing characteristics. For example, the Urtecho promoter dataset only utilizes a small number of different motifs (see Table below) and their combinations, which is not exhaustive; however, their experimental test system does a good job of measuring transcription rates (compared to prior studies) as they utilize a ribozyme to insulate transcription from translation and they integrate their test system into the genome to greatly reduce copy number variation. We also utilize our prior *in vivo* dataset (from Hossain et al.) containing 4350 highly non-repetitive promoter sequences, which are expressed from a multi-copy plasmid and do not use a ribozyme. We observe much higher variation in transcription rates from the Hossain dataset, which is not surprising given the copy number variation and confounding effects from differences in mRNA decay & translation. The Lagator et al. datasets (Pr and Pl) are constructed from randomized oligonucleotides, starting from a baseline sequence. As you can see from the Table below, when using that mutagenesis approach, the number of unique sequences per motif is directly proportional to the motif length. In contrast to our rational design approach, **the datasets in Lagator et al. do not broadly sample the -10 and -35 hexamer space.** Altogether, there is no such thing as a “perfect dataset” and it is essential to test model predictions on many datasets to critically test the model’s generalizability, which is what we have done in our manuscript.

Number of Unique Motifs	Tested by Urtecho	Tested by Lagator (Pl dataset) *	Tested by Lagator (Pr dataset) *	Tested by LaFleur
UP	2	1738	6463	605
-35 hexamer	8	202	278	4096 (all possible)
-10 hexamer	8	224	262	4096 (all possible)
Discriminator	8	271	313	735
Spacer	8	1501	3597	229
-10 extended	8	110	109	256 (all possible)
ITR	8	578	562	582
* From Lagator et al. The random P_R mutant library was created by cloning custom-made oligonucleotides (IDT Technologies), which had a 12% mutation rate for each of the 67 positions in the P_R promoter (4% mutation chance for each possible mutation away from the wildtype). The P_L mutant library was created in the same way except using mutation rates that were 9% per nucleotide (with 3% mutation chance for every possible non-wildtype mutation).				

As part of our model development, and with the results of Einav & Phillips 2019 in mind, in our original submission, we developed and tested a non-linear free energy model that includes all possible forms of cooperativity or anti-cooperativity that could take place between pairs of motifs. We apologize for not doing a great job of explaining this non-linear model in the original manuscript text and highlighting its comparison to the linear model. **In the revised manuscript, we have added a section to the Results that more fully describes the development of the non-linear model (which we call the quadratic model), its training using the Urtecho dataset to identify the quadratic term coefficients, its accuracy metrics when applied on all datasets, and a thorough accuracy comparison with respect to the linear model.** We found that the quadratic model is more accurate on the Urtecho dataset (its training dataset), but it had the same/similar or lesser accuracy on all other datasets (when using Pearson R^2 as the metric) and it has lesser accuracy on all other datasets (when using the MAE or MSE as the metric). In the original submission, we offered an alternate explanation for the transcription inhibition trend observed in the Urtecho et al. data (**Figure S7**). In the revised manuscript, we expand the Discussion to provide more insight regarding this trend and highlight the lack of conservation across the other 31,000 *in vivo* promoters tested. Overall, there is not convincing data-driven evidence that avidity or cooperativity exists between the -10 and -35 hexamers. It is possible that the sigmoidal shape of the Urtecho dataset, which is caused by its genome-integration, low “floor” in measured transcription rates and barcoding strategy, is artificially supporting the usage of a non-linear model.

Furthermore, the focus on what goes on within each component while the context is fixed means that it is not clear from the presented manuscript that this model significantly outperforms the existing models. For example, the authors present a comparison to a recent paper by Lagator et al., and discuss that their model performs better on two independent datasets. However, their model performs very poorly on the datasets generated by that paper, and especially on the fully random mutant library (36N) where their model has an R^2 of 0.15. This poses a serious question about the generalizability of the model, which is not helped by the fact that the authors do not discuss the libraries in which their model performs poorly but rather focus only on those examples where it performs better.

When the MAE or MSE metric is used, it becomes clear that our model predictions have the same or higher accuracy on all datasets, including the PI, Pr, and 36N datasets collected in Lagator et al (Table S4). The Pearson R^2 metric is notoriously sensitive to clustering at the extremes of the range, which is observed in the PI and Pr datasets. Moreover, it is expected that the Lagator model performs well on its own training data, but as highlighted above, their model training is highly biased to the 36N dataset. The high R^2 on the Pr and PI datasets come from clustering of predictions at the extreme end (Figure S11, Figure S13). When looking at completely unseen test data (Urtecho et al. and Hossain et al.), their extended model underperforms when compared to the standard model. This brings into question the motivation for their proposed extended model, and supports a linear additive model being the superior choice.

I suspect that the two issues – of novelty and of generalizability – are linked, because the outstanding and important advances in the prediction of promoter function I believe stem from understanding how different components of a model interact with each other. Together, these concerns potentially make the audience of this paper more specialized, as the novelty is contained to synthetic biology applications (Fig.5 and 6) and the evaluation of less important contributors to expression levels (like the ITR and the UP element) (Fig.1D and 2C).

We hope that we have clarified the novelties of our work and the generalizability of our model's predictions. Our dataset is highly novel & avoids many of the confounding interactions & sources of variation found in other datasets. Our model is more accurate than prior models. And we apply our model to design synthetic promoters with targeted transcriptions and to remove internal promoters from other genetic parts. We show that the UP and ITR motifs play an important role in controlling transcription rate and are necessary interactions to include as we do include in our model (unlike other models). We show that most core promoters used in Synthetic Biology are affected by genetic context effects (avoidable using our model predictions & design). We show an example where internal promoters are found in an engineered genetic system, which can be removed by leveraging our model predictions & design algorithm.

Detailed comments:

It would be very useful to provide a breakdown of MSE of the presented model based on the general expression level being predicted. What I mean is, what is the MSE for promoters that have high expression level, vs those with intermediate, vs those with low. This would help understand if the model performs equally well across the entire expression range, or if it underperforms at some expression level ranges. If the later is the case, it would be important to explore why that might be, especially in the light of various mechanisms of promoter function not included in this model, such as those explored by Einav and Phillips or Lagator et al.

We fully agree that the MSE (or MAE) is a better metric for accuracy and that it is important to test accuracy across the range of predicted transcription rates. **To answer the reviewer's question, we calculated the MAE and MSE for datapoints within binned predicted transcription rates for each dataset.** We show this analysis in Figure S4 and we discuss this analysis in the Discussion section. The results show prediction accuracy was highest when there are a larger number of datapoints within each bin, suggesting that all datasets did not uniformly sample the transcription space.

Figure S4: Model Residuals Across Transcription Space. (A) LaFleur linear model residual distributions across 20 evenly spaced binned binding free energies for Urtecho et al. (left), Hossain et al. (middle) and LaFleur et al. (right). **(B)** LaFleur linear model residual distributions across 20 evenly spaced binned binding free energies for the PI dataset from Lagator et al. (left), Pr dataset from Lagator et al. (middle) and 36N dataset from Lagator et al. (right). Blue lines are MAE values which correspond to the left y-axis, red lines are the number of variants in each bin which correspond to the right y-axis.

It would be really important to understand why the presented model performs poorly on the fully random promoter library from Lagator et al. (the 36N library). The low performance on random sequences strongly suggests that complex interactions based on genetic context of the promoter play a large role, at least while a promoter is defined as leading to a single TSS. Fig.5 addresses some of these questions, but in a very limited way using only 16 mutants. Providing deeper insights into what happens when a promoter moves from a given mutational neighbourhood to which their model is optimized and into random sequence space would be critical to increase the novelty of this work.

We agree with the reviewer that the 36N promoter library remains challenging to predict its transcription rates. Most of the dataset contains weak promoters and our model calculations show that there are many predicted transcriptional start sites with low transcription rates. Further, **Figure S4** (copied above) highlights that the 36N dataset lacks sufficient coverage of strong promoters, which could artificially reduce R^2 since the full range of transcription space is not sampled. As mentioned previously, our model still achieves lower MSE and MAE (higher accuracy) on this dataset as compared to the Lagator et al. model. But from a practical point of view, we would never design a promoter sequence with so many transcriptional start sites as it becomes an unnecessary confounding variable to controlling transcription rate.

Another good test in Fig.5 would be to design more promoters predicted to have same expression levels but very divergent sequences, to evaluate how repeatable the predictions are.

We agree with the reviewer and have carried out additional experiments to answer this question. **We designed and characterized an additional 19 non-natural promoters with highly**

dissimilar sequences with single TSSs and targeted transcription rates that span the accessible range, increasing the total count to 35 promoters, and increasing the dynamic range to 1525-fold. Designs include promoters of shorter length, which rely on the model's ability to accurately quantify the effect of the context DNA in forming the full-length promoter. This data is now included in **Figure 5**.

Legends to some of the figures and tables could be expanded to give more details. For example, Tables S4,5,6 – what do the colours indicate?

We have expanded the legends of all Supplementary Figures and Tables S4, S5, and S6 to provide more detail. Green backgrounds in Tables S4, S5, and S6 indicate datasets which are unseen in training for both the LaFleur et al. and Lagator et al. models. The grey and white background colors in Tables S4, S5, and S6 have no meaning except to group together the statistics for the same datasets.

I think one of the biggest contributions of this paper is how they used ML to derive model parameters that were comprehensible mechanistically. Expanding a bit in the main text on how this was done would be useful.

Yes, we agree that the model's physically meaningful coefficients are an important contribution. In the original manuscript, we provided an extensive explanation of the methodology behind this in the Methods. To further highlight the novelty of the approach, we added a paragraph to the Discussion section.

In the -10 binding motif, the authors examine the role of mutations by looking at every possible 3-mer. Energy Matrix-based models look at each individual position. I am not sure what the difference is between these two ways of modelling binding. For example, does the 'individual' position approach linearly add up to the observed effects of 3-mers? If not, when does it step away and why? Doing such an analysis can result in key insights into the interactions between mutations within each promoter component.

In our model building process, we trialed several different approaches to describing the hexamer interactions, including models that depended on each 1-nt position (mono-nt models), each pair of 2-positions (di-nt models), three adjacent 2mers (3 x 2mers), and two adjacent 3mers (2 x 3mers). We found that the two adjacent 3mer model yielded the highest accuracy and recapitulated known canonical motifs. The 2x3mer model fully encompasses the mono-nt model, but also includes synergistic interactions within each 3mer. We have included analyses in the revised Discussion section that compare the mono-nt vs. tri-nt models on the Hossain et al. and Urtecho et al. datasets, as well as a comparison on when the mono-nt and tri-nt models diverge (**Figure S8 and Supplementary Data 4**).

What is the distribution of spacer lengths in the experimental dataset, are they evenly distributed across all 5 relevant lengths?

There are 229 characterized promoter sequences with systematically varied spacer lengths and spacer nucleotide compositions. When applying our statistical thermodynamic model using the free energy coefficients, we enumerate all possible RNAP/ σ^{70} binding sites across a range of spacer lengths. We found that promoters with highly non-optimal spacer lengths (more than 20 nt, less than 15 nt) have much higher binding free energies (less favorable) as compared to another potential binding site in the promoter (even when that alternative site is using non-canonical -35 and -10 hexamers).

How long are the UP and ITR sequences used in the ‘Promoter Context Effects’ section? It’s possible that their relatively low effect on model accuracy is due to their short length, meaning that what ‘context’ is might be longer than the varied UP and ITR elements. Therefore, it would be good to note their exact length.

The UP and ITR sequences are 30 bp long. We have edited the manuscript to include this.

The introduction mentions the word ‘interactions’ quite a bit, but as discussed above, it is unclear which interactions are actually studied. In particular, given that the experimental design has a very limited exploration of the interactions between promoter components (what I think is the most important area where we lack knowledge), I would avoid using this word.

We use the word ‘interactions’ to describe the physical contacts between the amino acid residues in RNAP/ σ^{70} and the DNA nucleotides in the promoter. When RNAP/ σ^{70} has bound to a site, there are also entropic changes (e.g. stretching or compression of RNAP/ σ^{70} and kinking of the double-stranded DNA). Altogether, these interactions control the quantity of Gibbs free energy released when RNAP/ σ^{70} binds to a site.

Authors claim ‘a key novelty of our model is its complete set of interaction energies, covering all possible promoter sequences...’ – It would be important to clarify what the word ‘interaction’ means here, because it applies to only a small number of potential interactions between promoter components.

We’ve clarified this sentence to be specific to the interactions between RNAP/ σ^{70} and double-stranded DNA.

The ‘model generality’ section would be easier to follow if the authors referred to the external datasets by the author names, as opposed to just the reference, as it is difficult to go back and forth to the reference section to recall that 16 is Urtecho et al. and 12 is Hossain et al. etc.

We have revised the manuscript to refer to these datasets by their first authors.

Reviewer #2 (Remarks to the Author):

In the manuscript by LaFleur et al. (“Automated Model-Predictive Design of Synthetic Promoters to Control Transcriptional Profiles in Bacteria”), the authors create a biophysical model of transcription initiation of sigma -70 promoters in E. coli and train their model using high throughput assays and machine learning. Starting with 472 biophysical parameters, they train and

test their model on 5193 synthetic promoter variants and use the resulting data to prune their model to 346 parameters. With this model, they characterize the impacts of individual promoter motifs on transcription initiation. The authors test their model against a previously published model using their own in vitro dataset, as well as the Urtecho and Hossain datasets. The authors show that the new model yields significant improvements in predicting transcription rates for synthetic promoters ($R^2 = .79, .60, \text{ and } .45$ respectively for the new model; vs. $R^2 = .60, .45, \text{ and } .39$ for previously published model). They also demonstrate the applicability of their model for gene circuit debugging and the generation of de novo promoters with desired transcription rates. This model, accessible as an online software tool, is well-suited for synthetic biologists looking to reliably predict and control transcription rates of their genetic constructs. This work advances the field of genetic circuit design by combining a biophysical model for transcription initiation with machine learning informed by massively parallel in vitro assays. This is a growing area of research and this is nice work that advances the state-of-the-art. The writing and figures are well composed and easy to understand. One the following comments are addressed, it should be suitable for publication in Nature Communications.

Major Comments:

1. The authors seem to sweep under the rug the fact that of their initial 14206 promoters, only 5193 are used due to the recurring presence of a cryptic TSS. It would be helpful for the authors to explain the impact of the cryptic promoter within the ORF more explicitly. According to Fig 1C, this cryptic TSS looks to be consistently around position 105. Would the inclusion of promoters with cryptic start sites in the model training help inform parameters and motifs that more likely lead to a cryptic TSS? Could this improve biological relevance for predicting transcription rates of endogenous promoters and promoter design?

We apologize for presenting a potentially misleading impression. The 5193 promoter sequences were selected because they have a single predominant transcriptional start site where the promoter's motifs could be conclusively identified. This is important during model training to avoid incorrectly assigning a transcription rate measurement to an incorrect labeling of motifs. Otherwise, without the transcriptional start site mapping and filtering, the model would be trained on incorrectly labeled data. It is notable that our work is the first to apply transcriptional start site mapping to training predictive models of transcription rate. Prior models simply assumed that the transcriptional start site never shifted, which is incorrect. Our usage of TSS mapping and data filtering during model training does improve model accuracy, which is seen in the benchmarking metrics & statistics across all datasets.

2. Additionally, what is the composition of the 5193 promoters used? Is the set skewed to variations within one of the promoter regions? It seems like this could skew the retained features and decrease the generalizability of the model. For instance, Table S1 shows all -10 and -35 3-mers were retained, but no ITR features were retained. What is the explanation for this, and how are the Gibbs free energy contributions calculated when no features are retained?

The 5193 promoter sequences contained sequences encompassing all of the 472 features (interactions) originally proposed to be included in the model. The filtered data did indeed have sufficient information to distinguish the effects of these interactions. However, during model training & feature reduction, we found that the presence of 126 features (including several

hypothesized interactions affecting ITR function) did not improve the model's accuracy on the test dataset, were correlated with other more important features, or had the effect of slightly skewing 3mer coefficients away from known canonical interactions (ie, our "positive controls"). By removing these 126 features, we arrived at a model with high accuracy on both the training and test datasets and that recapitulated the known canonical interactions. In the methods section in the original manuscript, we defined each of the retained features, which included the R-loop strength calculated from the ITR sequence. All promoter regions retained at least one feature.

3. Regarding the model, the connection between the 346 features to the different Gibbs free energy contributions is not clear, especially the categorical properties such as spacer length. This information seems crucial for a biophysical model and should be included to strengthen the manuscript. Additionally, it is not obvious how to interpret and compare the different interaction energies in Figure 2E, have they been normalized? For instance, is a strong -35 interaction equal to a strong -10 interaction?

Our Methods section describes how the machine learning was carried out to identify the interaction energy coefficients corresponding to each feature/interaction. We expanded our Methods to include details how we processed numerical and categorical features during training. Importantly, all of the coefficients have values on the same energy scale (in units of RT). In our free energy model, if the terms ΔG_{-35} and ΔG_{-10} both have the same energetic contribution (e.g. -1.0 RT), then they both contribute equally to determining the transcription initiation rate of the promoter.

4. The authors tried a linear combination of different features to estimate interaction energies, and a linear combination of these interactions energies to estimate transcription rates. The manuscript would be enhanced by rationale for why this approach was chosen and a more in-depth discussion of the limitations when compared to other approaches such as non-linear regression or neural networks.

As a comparison, we developed a non-linear model that includes cooperative/anti-cooperative coefficients for each pair of motif interactions. We briefly mentioned this model in the original manuscript submission, but it was not described well. In the revised manuscript, we have added a new section in the Results to explain how the non-linear model (with quadratic terms) was formulated and trained using the Urtecho dataset. We also included a discussion on the benefits of our method when compared to non-interpretable machine learning approaches. As a significant result, we found that the non-linear quadratic model does achieve higher accuracy on the Urtecho dataset, but its accuracy (MAE and MSE, **Table S4**) on all other data is either the same as the linear model or less. It is possible that the sigmoidal appearance of the Urtecho dataset, caused by the barcoding strategy and the "floor" in the observed transcription rate measurements, has the effect of incorrectly supporting the presence of cooperative interactions between the -10 and -35 hexamer contacts. Recall that the Urtecho dataset uses genome-integrated expression cassettes.

5. The authors do a good job by benchmarking different models and datasets, but it is hard to interpret the comparisons without stronger statistical tests, especially given the different number of parameters across models. The use of F-test or information criterion (AIC or BIC) to compare the models would help strengthen the author's claims.

In the revised manuscript, we carried out F-tests for each model comparison and included F-statistics and p-values in the Supplementary Data. The F-test tests whether our model achieves a statistically significant reduction in the variance of the model error distribution. We found that our linear model achieves lower model error variances (fewer outliers) as compared to Lagator et al.'s Extended model. We show the model error distributions in the Supplementary Information (**Figure S13**) and include the F-statistics in **Supplementary Data 4**.

6. Page 9, “However, most promoters do not contain canonical motif sequences; their TX rates are controlled by a mixture of weaker interactions. A key novelty of our model is its complete set of interaction energies, covering all possible promoter sequences, which provides the ability to predict the TX rate of any σ^{70} promoter.” The claim that their model covers “all possible promoter sequences” is misleading, especially given that only $\frac{1}{3}$ of designed promoters were used in the training of the model. Additionally, other models (Urtecho et al.) can also predict transcription rate of any σ^{70} promoter.

Given any DNA sequence, our model calculates the strengths of the interactions controlling how well RNAP/ σ^{70} binds to the DNA to initiate transcription. For each potential start site in the sequence, our model enumerates all potential binding sites, and calculates the transcription initiation rate of the most stable binding complex. It is within this context that we conclude that our model can predict the transcription initiation rate of any DNA sequence. The size, breadth, and composition of our training dataset was sufficient to encompass all of the interactions in the model.

Importantly, the Urtecho dataset includes only a very small number of motif sequences (e.g. only 8 different -10 hexamers and only 8 different -35 hexamers) used in combinations to create a set of promoters. The model described in Urtecho et al. is not a sequence-complete model. It can not accept an input of arbitrary DNA sequence and perform a prediction. Instead, the model expects the input to be restricted to the alphabet used in the training data (e.g. 1 out of 8 -10 hexamers, 1 out of 8 -35 hexamers).

7. While the exclusion of alternative sigma-factors in the *in vitro* assay helps illuminate the “true” transcription rate due to sigma70-RNAP interactions, it does come at the expense of biological relevance and potentially limits the generalizability of this model for the prediction of transcription rates and promoter design. This is shown by the model’s low accuracy on *in vivo* datasets. This limitation should be discussed more explicitly.

We agree that our model is currently limited to predicting sigma70-dependent promoter transcription rates. However, we do validate our model predictions on *in vivo* datasets (grown in the exponential phase of growth where sigma70 is predominant) and we do find that our model is more accurate than other models (Lagator) that were trained using only *in vivo* measurements. We conclude that our *in vitro* dataset provides more precise measurements that avoid several of the confounding factors present in *in vivo* datasets, leading to higher overall accuracy & generalizability.

Further, all models developed to date have only ever been tested on cells growing in the exponential phase of growth. If we would like our models to predict transcription rates when cells are grown in other phases (e.g. stationary or heat shock), then we can not create a single

model that mixes together the effects of multiple sigma factors. Instead, we need to develop multiple sigma-specific models and then combine together their predictions coincident with the sigma factor levels in each growth condition. Our current work shows that developing a sigma-specific model of transcription initiation rate is not only possible, but advantageous over prior approaches. In ongoing work to be submitted in the future, we will describe additional sigma-specific models that encompass multiple growth conditions.

Minor Comments:

1. Discussion about challenges to generalizing this work to other organisms and other sigma-factors would be valuable.

We have revised our Discussion section to highlight how our approach could be extended to additional sigma-factors and organisms. In particular, our *in vitro* approach makes it possible to develop predictive models of transcriptional initiation even when the organism remains genetically recalcitrant (e.g. low transformation efficiencies).

2. In Fig 1D, the range transcription rates of each motif library is shown. The “all” bar may be misleading, as this represents the greatest difference between a single variant in one motif and a single variant in another motif. A total range of 133 does not cover the wide range of transcription rates in the genome. The model may behave poorly at the extremes (very low transcription or very high transcription), and there is no data or much discussion to address this point.

The reviewer has brought up an interesting point. The range of transcription rates shown in Figure 1D are specific to constitutive promoters without any other factors present. Natural endogenous promoters will also be regulated by repressor and activator transcription factors that will decrease/increase their transcription rates by even greater amounts. There are also chromatin-like proteins (e.g. Fis, IHF, H-NS) that non-specifically regulate promoter transcription rates. In particular, we anticipate the weakest constitutive promoter can be greatly repressed by factors, resulting in a much lower transcription rate (reducing the minimum in the range down towards zero). We have revised the Discussion section to explicitly clarify this point.

To address model accuracy across the range of transcription rates, we have calculated the MAE across promoters within binned transcription rates spanning the range (see plot below). This analysis is shown in **Figure S4** and reported in the Discussion section.

Figure S4: Model Residuals Across Transcription Space. (A) LaFleur linear model residual distributions across 20 evenly spaced binned binding free energies for Urtecho et al. (left), Hossain et al. (middle) and LaFleur et al. (right). (B) LaFleur linear model residual distributions across 20 evenly spaced binned binding free energies for the PI dataset from Lagator et al. (left), Pr dataset from Lagator et al. (middle) and 36N dataset from Lagator et al. (right). Blue lines are MAE values which correspond to the left y-axis, red lines are the number of variants in each bin which correspond to the right y-axis.

3. “We next evaluated whether these genetic context effects are potentially widespread by carrying out Monte Carlo simulations to predict the range of TX rates expected when varying the UP and ITR regions of several commonly used promoters (Figure 4B). Promoters with lower average TX rates exhibited higher sensitivity to changing the UP and ITR regions with an overall average coefficient of variation of 0.40.” Cannot tell this statement is true based on Figure 4B, there is no statistical significance test. Moreover, there is no methods section for Monte Carlo simulations.

We carried out T-tests to find that promoters exhibited similar sensitivity to changing the 30 bp UP and ITR regions regardless of whether those promoters had high or low transcription rates ($p = 0.06$). However, when the context length is increased to 50bp, the sensitivity does become statistically significant ($p = 1.8e-16$). We have updated this section of the manuscript text to reflect these findings. We have also expanded the Methods section to include the Monte Carlo simulations.

4. Page 5, "Here, we arbitrarily set $DG_{total,ref}$ to zero so that stronger (weaker) interaction free energies have more negative (positive) values in comparison." This sentence is hard to follow.

We revised the manuscript to clarify this sentence. The free energy scale is defined so that our reference promoter has a $\Delta G_{total,ref}$ of zero. The choice is arbitrary and does not affect any of the accuracy metrics. But selecting a zero reference energy means that a weaker than average interaction is associated with a positive free energy value and a stronger than average interaction is associated with a negative free energy value.

5. Readability of Fig 6 is difficult. Axes for 6a can be standardized. Putative start sites should be labeled. Additionally, it is unclear where the cut off for a TSS comes from in 6a.

The y-axes in Figure 6A show predicted transcription rates and measured flux ratios. The scales are different because transcription rates are predicted on one scale and flux ratios are measured on another scale. Rather than normalizing and obscuring the origins of the numbers, we purposefully plotted the raw predictions and the raw measurements even though they use two different scales. The horizontal dashed lines show the cutoff values for the definition of a TSS. We have clarified this in the figure legend. Experimental TSS cutoffs are described Borujeni et al., and predicted TSS cutoffs in the Methods section.

6. How does a 55% prediction accuracy for identifying cryptic promoters compare to other methods?

This is somewhat an apples-to-oranges comparison as commonly used promoter finders will only report the presence/absence of a promoter with some score threshold. We ran BPROM (<http://www.softberry.com/berry.phtml?topic=bprom&group=programs&subgroup=gfindb>) on the circuit DNA sequence for an analogous comparison. The BPROM algorithm found only 16 promoters (on both strands), which is much fewer than the 51 cryptic promoters (both strands) that our algorithm identified. The 16 promoters identified by BPROM contained canonical-like hexamer motifs whereas our model was able to identify cryptic promoters that utilized non-canonical motif sequences.

7. Why is the train-test split 90-10 when figure 2A shows minimum error at ~25% training set size?

Figure 2A shows that model's mean absolute error converge to the same number when utilizing either the train dataset or the test dataset. This point of convergence occurs when only about 1250 datapoints are utilized, which is much less the total size of the dataset. This is called a learning curve in the machine learning field. It shows that model training did not over-fit the data and that we had sufficient data to develop a generalized model. If the mean error on the training dataset was always less than the mean error on the test dataset, that would indicate over-fitting. And if the mean errors across the training & testing datasets never converged, that would mean that we did not have sufficient data to fully train the model.

Reviewers' Comments:

Reviewer #1:

Remarks to the Author:

I am very satisfied with how the authors have addressed the comments. In particular, I think the introductions to the manuscript that the authors did to emphasize the novelty of their manuscript are excellent and will help increase the readership of the manuscript.

Reviewer #2:

Remarks to the Author:

We reviewed the revised manuscript by LaFleur et al. ("Automated Model-Predictive Design of Synthetic Promoters to Control Transcriptional Profiles in Bacteria"). The revised manuscript is notably improved compared to the initial submission. Most substantively, additional comparisons, statistical tests and discussion related to the models are now included. Overall, this reviewer's concerns have been addressed.